# WatchLog: From a Glimpse to Decision—Rapid Event Reasoning in Endpoint Detection and Response Logs with Multimodal LLMs

## Abstract

Endpoint Detection and Response (EDR) systems are essential for detecting malicious activities on endpoint devices, yet existing approaches struggle to efficiently process ultra-long log sequences and provide interpretable reasonings for security analysts. This paper presents **WatchLog**, a framework that models raw logs as video-structured representations to enable efficient video-language modeling of endpoint behaviors. Specifically, each event is encoded into a key-value guided image, and the resulting images are temporally arranged into a video-structured sequence. A temporal cross-attention mechanism then performs pixel-wise temporal aggregation, producing compact sequence embeddings that preserve behavioral fidelity while reducing computational cost. We conduct two-stage pre-training followed by supervised fine-tuning to generate behavioral explanations grounded in the semantics of event sequences and final judgments. Experiments on our newly constructed EDR8M-20R dataset demonstrate that WatchLog achieves higher detection accuracy and recall than the state-of-the-art baselines, while also generating reliable reasoning traces and enabling more efficient inference. Furthermore, our real-world application of WatchLog has validated its efficiency, effectiveness, and strong generalization capabilities.

## 1 Introduction

As cyberattacks become increasingly stealthy and sophisticated Kaur & Ramkumar (2022), Endpoint Detection and Response (EDR) systems have become indispensable for continuous monitoring and forensic analysis on endpoint devices Hassan et al. (2020); Dong et al. (2023); Sharif et al. (2024). Although EDR logs capture rich behavioral traces, their scale and complexity pose unique challenges. First, a single log may include thousands to tens of thousands of events—each with long textual descriptions—yet only a few are security-relevant. Second, malicious actions are temporally sparse and often mimic benign behavior, making them difficult to identify. Third, real-world security workflows necessitate not only accurate detection but also interpretable reasoning to enable effective decision-making by security analysts.

These challenges expose the limitations of existing approaches. Most deep learning methods lack explicit mechanisms for temporal reasoning and fail to provide interpretability aligned with human analytic needs. Directly applying LLMs to raw logs faces scalability and efficiency bottlenecks, primarily due to two factors: (*i*) the ultra-long and high-dimensional nature of raw logs often exceeds the context windows of practical LLMs; (*ii*) malicious behaviors are temporally sparse and hidden among numerous noisy events, making stealthy attack trajectories difficult to uncover.

To overcome these limitations, we introduce **WatchLog**, a three-stage framework that progressively transforms raw logs into semantic embeddings interpretable by multimodal LLMs, ultimately producing behavioral judgments with explanatory rationales, as illustrated in Figure 1. Specifically, in the **first stage**, we introduce a novel event-to-image transformation that encodes key-value fields into pixel-level embeddings while preserving field-level semantics. An image-event alignment pre-training then aligns the visual encoder with event-level textual semantics. In the **second stage**, event-level images are stacked into a video, and an additional temporal cross-attention block is introduced to capture cross-event dependencies. This produces compact sequential embeddings that

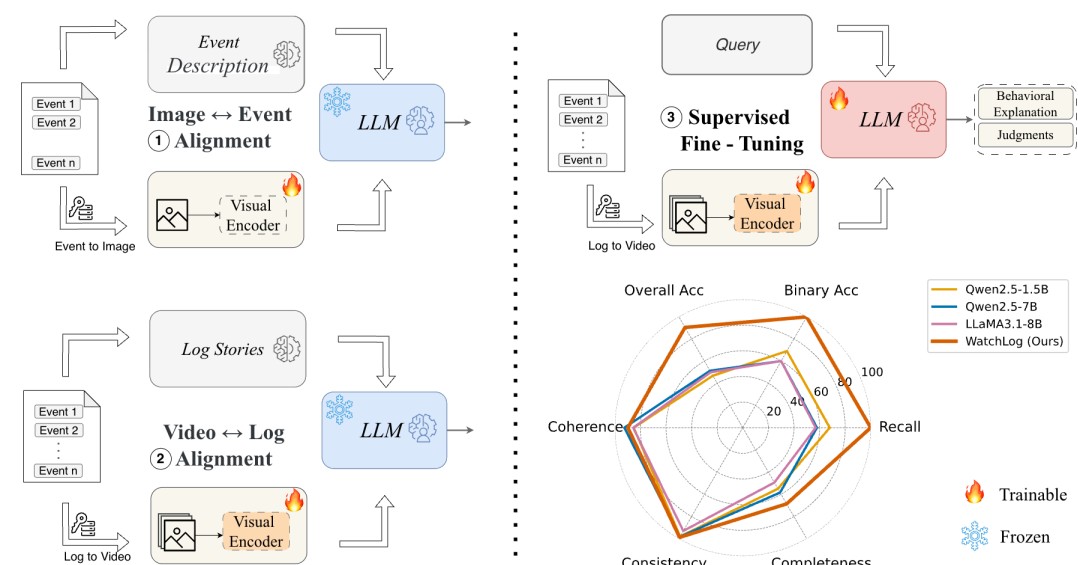

Figure 1: Overview of WatchLog Framework with Representative Results

preserve global semantics while mitigating the cost of long-context processing. A video-log alignment pre-training further ensures consistency with holistic log semantics. In the **third stage**, the compact representations are fused with a trainable LLM fine-tuned under a reasoning-aware objective, enabling WatchLog to generate threat predictions with their associated explanations. Representative comparisons against LLM baselines are shown in the bottom-right radar chart of Figure 1, with detailed numerical results provided in Table 2. We evaluate our framework from two perspectives: recognition accuracy, measured by exact match with ground-truth family names, and rationale quality, assessed by GPT-5 for coherence, consistency, and completeness.

To the best of our knowledge, this is the first work to transform EDR logs into video-structured inputs and leverage a multimodal LLM to perform reasoning, bridging security log analysis with visual-language modeling. Our main contributions are as follows:

- We present WatchLog, a novel multimodal paradigm for EDR log analysis, enabling end-to-end reasoning for accurate behavioral judgment and rationale generation.
- We propose a spatio-temporal modeling scheme that encodes events as images and introduces temporal cross-attention to generate compact sequential embeddings, preserving both field-level and global semantics while reducing computational overhead.
- We design a progressive three-stage training strategy, including image-event and video-log pre-training to align visual and textual semantics at both event and log levels, followed by supervised fine-tuning for adaptation to the downstream task.
- We demonstrate the effectiveness and efficiency of WatchLog through comprehensive experiments, showing consistent improvements in threat detection and rationale generation, and strong generalization under both closed-set and open-set evaluation scenarios.

## 2 RELATED WORK

Endpoint Detection and Response (EDR) systems enable enterprises to identify potential security threats by continuously monitoring the activity of the system on endpoint devices, including computers, servers, and mobile devices. This section reviews relevant EDR solutions and associated technologies, providing valuable insights for our research.

**Machine Learning Methods.**

Traditional endpoint threat detection initially relied on rule-based signatures Hassan et al. (2020), which offered limited adaptability to novel or evolving attacks. Classical machine learning (ML)

approaches improved this adaptability by extracting hand-crafted features from logs or files and training classifiers Najafi et al. (2024). For example, WATSON Zeng et al. (2021) employed IDF and hierarchical clustering to derive semantic representations of events, while Kumar et al. (2022) utilized image-based texture features with k-NN and Naïve Bayes classifiers. However, these methods are highly dependent on domain-specific feature engineering, and static feature representations often fail to capture the evolving semantics of event sequences, limiting their effectiveness in detecting sophisticated and stealthy threats.

**Deep Learning Methods.**

Deep learning (DL) enables automatic feature extraction, allowing models to learn attack patterns directly from raw logs Yu et al. (2019). LSTM-based approaches Ring et al. (2021) captured sequential dependencies, while graph-based methods such as ProGraPher Yang et al. (2023) integrated graph embeddings with sequence models to improve detection accuracy. Recent works further exploited adversarial debiasing Tsai et al. (2024) or self-supervised pre-training Sharif et al. (2024). Despite these advances, most DL approaches still treat logs as either long unstructured text or simple sequential data, lacking explicit event-level structure. As a result, they are limited in capturing salient semantic features of individual events and modeling the temporal evolution of log sequences, which motivates our spatio-temporal modeling approach for EDR log analysis.

**LLM-based Approaches.**

Large Language Models (LLMs) Brown et al. (2020) have recently advanced tasks such as malware analysis, phishing detection, and intrusion prevention Li et al. (2024); Hu et al. (2024); Sun et al. (2024); Zhou et al. (2025). Compared with DL methods, LLMs demonstrate stronger capabilities in semantic understanding and analogical reasoning Snell et al. (2025), making them promising for detecting evasive behaviors in noisy logs. For example, Mal-LLM Xue et al. (2024) used chain-of-thought prompting to guide LLMs in capturing the semantic intent of malicious code alongside classical ML classifiers. AppPoet Zhao et al. (2025) employed LLM to generate rich linguistic descriptions of software data, thereby improving both detection accuracy and interpretability. However, directly applying LLMs to raw EDR logs remains problematic: log sequences can exceed hundreds of thousands of tokens, posing challenges for context retention and efficient inference. Moreover, the sparse distribution of malicious events Dong et al. (2023) and their high similarity to benign operations Sim & Borthwick (2018) further increase the reasoning complexity in long contexts. Unlike existing methods, we propose a novel multimodal framework that effectively leverages LLMs' semantic understanding and reasoning capabilities to detect and interpret malicious behavior.

**Long-Context Understanding in LLMs.**

A parallel line of research seeks to extend the context length of LLMs, including RoPE-based extrapolation (YaRN) Peng et al. (2024), bi-level attention construction (SelfExtend) Jin et al. (2024), and progressive interpolation strategies (LongRoPE) Ding et al. (2024). While these approaches expand the context window to millions of tokens, they remain computationally expensive and are ill-suited for the ultra-long and high-volume sequences typical of EDR data. Instead of relying on sheer context extension, our method transforms logs into structured images and video-like temporal representations for multimodal LLM processing, analogous to video comprehension where salient information is selectively attended to. This approach both improves computational efficiency and preserves the semantic precision of events and the temporal dynamics of sequences, providing a more practical paradigm for reasoning-aware malicious behavior detection.

## 3 METHODS

We propose a three-stage framework for malicious behavior detection and reasoning on EDR logs. The core idea is to progressively transform raw logs into structured spatio-temporal representations that can be interpreted by multimodal language models, addressing challenges such as ultra-long sequences, noisy events, and the need for interpretable reasoning. In the first stage, individual events are converted into images and aligned with their textual descriptions; in the second stage, these event-level images are organized into a video-structured sequence and aligned with log-level descriptions; in the third stage, raw logs are converted into structured representations and fine-tuned under a supervised objective to generate behavioral judgments and explanatory rationales. This progressive design captures both event-level semantics and temporal dynamics, while leveraging

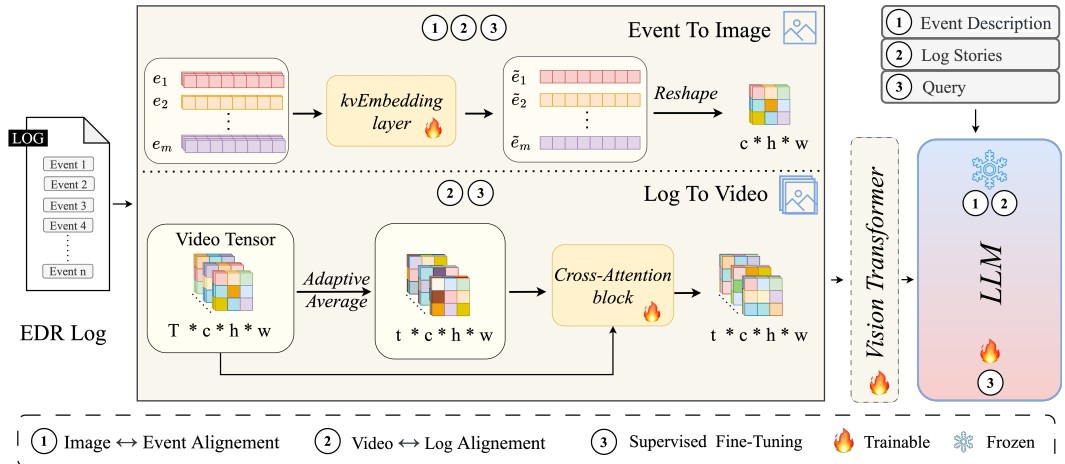

Figure 2: Overview of the **WatchLog** framework. In Stage 1, events are parsed and transformed into pixel embeddings via the kvEmbedding layer (MLP + softmax aggregation), and aligned with their textual descriptions through image-event pre-training. In Stage 2, images are temporally stacked and processed by an additional cross-attention block to produce compact video-like representations, which are further aligned with overall log semantics via video-log pre-training. In Stage 3, the unfrozen language model backbone is fine-tuned end-to-end together with the visual encoder to generate behavioral predictions and explanatory rationales. All stages share and fine-tune the same Vision Transformer (ViT) structure.

multimodal reasoning capabilities to achieve precise understanding of log behaviors. Appendix H provides a detailed analysis of the mathematical properties and computational complexity of each stage of our method.

## 3.1 IMAGE ↔ EVENT ALIGNMENT

In this stage, our goal is to construct structured event-level representations that support both image-event alignment and downstream modeling. Given a raw event $E$, we process it through two parallel pathways: (i) a *textual pathway*, where the event is reformulated by GPT Achiam et al. (2023) into detailed natural language descriptions $S$ (see Appendix Table 9 for prompt design); and (ii) a *visual pathway*, where the event is decomposed into key-value pairs $\{\mathbf{x}_m = (k_m, v_m)\}_{m=1}^M$, and each pair $\mathbf{x}_m$ is tokenized into embeddings $\{e_{m1}, \dots, e_{mL_m}\}$ with $e_{m\ell} \in \mathbb{R}^d$ and $L_m$ denoting its token length. To accommodate variable-length pairs, we introduce a *kvEmbedding* layer that applies a trainable projection with softmax-based weighting, yielding a fixed-dimensional embedding:

$$\tilde{e}_m = \sum_{\ell=1}^{L_m} \frac{\exp(W^\top e_{m\ell})}{\sum_{\ell=1}^{L_m} \exp(W^\top e_{m\ell})} \cdot e_{m\ell}, \tag{1}$$

with $W \in \mathbb{R}^d$, $d$ denoting the hidden dimension of the token embeddings. The resulting embeddings $\{\tilde{e}_1, \dots, \tilde{e}_M\}$ are then projected to the ViT hidden dimension $c$ and reshaped into an image-like tensor $\tilde{E} = \text{Reshape}([\tilde{e}_1; \dots; \tilde{e}_M]) \in \mathbb{R}^{c \times h \times w}$, with $M = hw$.

To align the two path, we perform image-to-description pre-training, where each event $E$ is processed to obtain an image $\tilde{E}$, which is aligned with its GPT-generated textual description $S$.

$$\mathcal{J}_{IEA} = -\sum_{l=1}^{L_I} \log P_{\theta'}(S_l \mid S_{<l}, E), \tag{2}$$

where $L_I$ denotes the number of tokens for the textual description $S$, $\theta'$ denotes all trainable parameters in this stage. This alignment grounds the visual representations in human-interpretable semantics, thereby enhancing their utility for subsequent log-level modeling and multimodal reasoning.

Table 1: Pseudocode of WatchLog under the proposed three-stage framework.

---

**Algorithm 1: WatchLog**

---

**Input:** raw log file $\mathcal{L}$ with $T$ events. Query $\mathbf{q}$.
**Output:** attack family label $y$ and reasoning description $r$.

**Stage 1: Image-Event Alignment**
   **for** event $E_i$ in log file $\mathcal{L}$ **do**
      decompose $E_i \rightarrow \{\mathbf{x}_m = (k_m, v_m)\}_{m=1}^{M}$
      tokenize $\mathbf{x}_m \rightarrow \{e_{m\ell}\}_{\ell=1}^{L_m}$
      transform kvEmbedding$(e_m) \rightarrow \tilde{e}_m$
      reshape Reshape$([\tilde{e}_1; \dots; \tilde{e}_M]) \rightarrow \tilde{E}_i$
      $\mathcal{J}_{IEA} = -\sum_{l=1}^{L_I} \log P_{\theta'}(S_l \mid S_{<l}, E_i)$
   **end for**

**Stage 2: Video-Log Alignment**
$\mathcal{L} \rightarrow \{\tilde{E}_i\}_{i=1}^{T} = \mathcal{E} \in \mathbb{R}^{T \times c \times h \times w}$
AdaptiveAvgPool$(\mathcal{E}) \rightarrow \mathcal{E}_{pool} \in \mathbb{R}^{t \times c \times h \times w}$
set $Q = \mathcal{E}_{pool}, K = \mathcal{E}, V = \mathcal{E}$
CrossAttn$(Q, K, V) \rightarrow \tilde{\mathcal{E}} \in \mathbb{R}^{t \times c \times h \times w}$
$\mathcal{J}_{VLA} = -\sum_{l=1}^{L_V} \log P_{\theta''}(S_l \mid S_{<l}, \mathcal{L})$

**Stage 3: Supervised Fine-Tuning**
   feed Query $\mathbf{q}$ and $\mathcal{L} \rightarrow \tilde{\mathcal{E}}$
   generate $y$ (attack family) and $r$ (reasoning)
   $\mathcal{J}_{SFT} = -\sum_{l=1}^{L} \log P_{\theta*}((y,r)_l \mid (y,r)_{<l}, (\mathcal{L}, \mathbf{q}))$

---

## 3.2 Video ↔ Log Alignment

In this stage, we extend the framework to temporal semantic alignment of logs. Given a log file $\mathcal{L}$ with $T$ events, we also process it through two parallel pathways: (i) a *textual pathway*, where the log sequence is reformulated by GPT Achiam et al. (2023) into detailed natural language descriptions (see Appendix Table 10 for prompt design); and (ii) a *visual pathway*, where the event-level images obtained from the previous stage are chronologically stacked into a video-like tensor $\mathcal{E} = [\tilde{E}_1, \tilde{E}_2, \dots, \tilde{E}_T] \in \mathbb{R}^{T \times c \times h \times w}$. To reduce redundancy while preserving salient temporal information, we introduce a temporal cross-attention block. Specifically, we first apply adaptive average pooling to generate a compact query set $Q = \mathcal{E}_{pool} \in \mathbb{R}^{t \times c \times h \times w}$ ($t \ll T$), while the original video-like embeddings serve as keys ($K$) and values ($V$). The cross-attention is then applied at each pixel location:

$$\tilde{\mathcal{E}}_{i,j} = \text{Softmax}\left(\frac{Q_{i,j}W_Q(K_{i,j}W_K)^\top}{\sqrt{c}}\right)(V_{i,j}W_V), \quad \tilde{\mathcal{E}}_{i,j} \in \mathbb{R}^{t \times c}, \tag{3}$$

where $K, V = \mathcal{E}$, $W_Q, W_K, W_V \in \mathbb{R}^{c \times c}$ are trainable projections, and $1 \leq i \leq h, 1 \leq j \leq w$ denote pixel coordinates on the image grid. This operation condenses the $T$ frames into $t$ abstracted frames while maintaining spatial consistency, yielding a video-level representation that highlights discriminative temporal dynamics.

To align the two modalities, we perform video-log pre-training, where each log $\mathcal{L}$ is processed to obtain the abstracted video $\tilde{\mathcal{E}}$, which is aligned with the GPT-generated log stories $\mathcal{S}$. We adopt an autoregressive objective: the multimodal model takes the original log $\mathcal{L}$ as input and generates the description tokens $\{\mathcal{S}_1, \dots, \mathcal{S}_{L_V}\}$, optimized by the negative log-likelihood loss.

$$\mathcal{J}_{VLA} = -\sum_{l=1}^{L_V} \log P_{\theta''}(\mathcal{S}_l \mid \mathcal{S}_{<l}, \mathcal{L}), \tag{4}$$

where $L_V$ denotes the number of tokens for the textual description $\mathcal{S}$, $\theta''$ denotes all trainable parameters in this stage.

## 3.3 Supervised Fine-Tuning

Moving from pre-training to task-specific optimization, equipping the model to produce both categorical predictions and human-interpretable rationale. The inputs consist of: (i) the raw logs $\mathcal{L}$, and (ii) an external query text $\mathbf{q}$, as shown in Appendix Table 11. The raw log inputs are processed through the same visual encoder as in the video-log alignment stage to obtain $\tilde{\mathcal{E}}$, and are fine-tuned end-to-end with the LLM under supervised objectives. Each training instance is paired with the ground-truth family label $y$, and a natural-language rationale $r$ that highlights the malicious evidence. We concatenate these into a unified autoregressive target sequence $\mathbf{y} = \{y_1, \dots, y_L\}$, where the family token appears first, followed by reasoning tokens. The model is trained by minimizing

the negative log-likelihood:

$$\mathcal{J}_{SFT} = -\sum_{l=1}^{L} \log P_{\theta^*}((y,r)_l \mid (y,r)_{<l}, (\mathcal{L}, \mathbf{q})), \tag{5}$$

where $\theta^*$ denotes all trainable parameters of the model. $L$ represents the total number of tokens in the ground-truth family and their corresponding rationales for the input log.

Importantly, under a fixed spatial resolution $(h \times w)$ and channel dimension $c$, the sequence length $T$ becomes the primary factor determining both time and memory costs, making temporal abstraction a key lever for scalability in practice.

## 4 EXPERIMENTS

### 4.1 EXPERIMENTAL SETUP

**Dataset.** We construct EDR8M-20R, the first large-scale well-annotated benchmark dataset tailored for end-to-end evaluation of EDR systems and conduct experimental evaluation on it. Compared with existing datasets Man Duc Trong et al. (2020); Zengy et al. (2022); Yang et al. (2023); Sharif et al. (2024), EDR8M-20R is a publicly released dataset that combines complete real-world endpoint execution traces with expert-verified reasoning paths and calibrated threat family annotations. The data collection and construction process is described in Appendix D. EDR8M-20R covers 20 behavior families (including normal behaviors), with 100 samples in each family. In our experiments, we randomly select 80 samples per family as the training set and reserve the remaining 20 samples for the test set. To further evaluate generalization, we construct two additional test sets: the within-distribution extended test set, which contains 1,000 additional samples from the same 20 families, and the out of distribution (OOD) test set, which consists of 112 samples from unseen families. In addition, to support the SFT stage, we leverage a large language model (LLM) with human verification to generate explanatory rationales for the behavior family of each sample. For OOD samples, the rationales indicate only that they are malicious, without disclosing their specific family labels to the model. The prompt design for rationale generation is provided in Appendix E.

**Baselines.** We compare against three representative traditional methods: LGTF Kumar et al. (2022), ADE Tsai et al. (2024), and DrSec Sharif et al. (2024); two general-purpose LLM baselines: DeepSeek-R1 Guo et al. (2025) and Qwen3-235B-A22B Yang et al. (2025); and other three LLMs with comparable sizes: Qwen2.5-3B, Qwen2.5-7B Yang et al. (2024), and LLaMA3.1-8B Grattafiori et al. (2024). Details are given in the Appendix F.

**Training Details.** We name our proposed model WatchLog, which is built upon the Qwen2-VL backbone model with approximately 2 billion parameters. In the alignment process between event and description, we replace the *pathEmbedding* layer of Qwen2-VL with a *kvEmbedding* layer, where the number of key-value pairs $M$ is statistically determined based on the training set and set to 144. For alignment between log stories and description, a cross-attention block is placed after the kvEmbedding layer, with the temporal compression dimension $t$ (default to 64). During training, the batch size and the epoch are set to 16 and 10. We use AdamW optimizer with an initial learning rate of $1 \times e^{-5}$ and a cosine annealing learning rate scheduler. We conducted all of our experiments on single node with 8 NVIDIA H100 (80G) GPU.

**Metrics.** For binary detection, we report *Binary Accuracy*, *Recall*, and *False Alarm*, which measure the overall correctness of distinguishing benign and malicious samples, the model's ability to detect malicious behaviors, and the proportion of benign samples incorrectly classified as malicious, respectively. For multi-family behavior detection, we adopt *Overall Accuracy*, which measures the proportion of all behavior families correctly identified. A prediction is considered correct only if the generated family name exactly matches the ground truth. For rationale evaluation, we assess the generated explanations along three dimensions: *Coherence*, *Consistency*, and *Completeness*. To reduce potential evaluation bias, we aggregate scores obtained independently from GPT-5, Gemini-2.5-Pro, and Qwen-3-235B-A22B, compute their average, and normalize the final result to a percentage scale. The full evaluation prompts are provided in Appendix G.

## 4.2 OVERALL PERFORMANCE

We compare WatchLog with three groups of baselines: (i) traditional EDR detection systems including LGTF Kumar et al. (2022), ADE Tsai et al. (2024), and DrSec Sharif et al. (2024); (ii) large-scale LLMs evaluated in a zero-shot setting (DeepSeek-R1 Guo et al. (2025), Qwen3-235B-A22B Yang et al. (2025)); and (iii) instruction-tuned mid-scale LLMs (Qwen2.5-1.5B, Qwen2.5-7B, and LLaMA3.1-8B). As shown in Table 2, WatchLog achieves the best overall performance across all metrics, with near-perfect detection (Binary Accuracy of 99.8%, Recall of 100%, False Alarm of only 5.0%) and superior reasoning quality, setting a new state of the art on EDR8M-20R.

Table 2: Comparison of detection performance (%) and reasoning quality (scaled to 100) on EDR8M-20R. $\diamond$ indicates zero-shot evaluation. ♠ denotes supervised fine-tuning (SFT). Reasoning metrics are only reported for models capable of generating rationales. All metrics are reported in percentage (%). FA = False Alarm, R = Recall, BA = Binary Accuracy, OA = Overall Accuracy, Conh. = Conherence, Cons. = Consistency, Comp. = Completeness.

| Method | Detection Perf. | | | | Reasoning Quality | | |
|---|---|---|---|---|---|---|---|
| | BA ↑ | R ↑ | FA ↓ | OA ↑ | Coh. ↑ | Cons. ↑ | Comp. ↑ |
| LGTF Kumar et al. (2022) | 92.8 | 96.0 | 65.0 | 48.8 | – | – | – |
| ADE Tsai et al. (2024) | 93.8 | 99.0 | 100.0 | 20.5 | – | – | – |
| DrSec Sharif et al. (2024) | 92.0 | 96.0 | 90.0 | 38.8 | – | – | – |
| DeepSeek-R1-671B $\diamond$ Guo et al. (2025) | 87.3 | 90.0 | 70.0 | 8.8 | – | – | – |
| Qwen3-235B-A22B$\diamond$ Yang et al. (2025) | 93.3 | 97.0 | 85.0 | 7.8 | – | – | – |
| Qwen2.5-1.5B♠ Yang et al. (2024) | 69.0 | 68.0 | 15.0 | 46.8 | 94.4 | 97.8 | 55.4 |
| Qwen2.5-7B♠ Yang et al. (2024) | 60.0 | 58.0 | **0.0** | 51.3 | **96.4** | **98.2** | 59.2 |
| LLaMA3.1-8B♠ Grattafiori et al. (2024) | 60.0 | 57.0 | **0.0** | 50.0 | 91.3 | 92.9 | 50.8 |
| **WatchLog-2B (Ours)** | **99.8** | **100.0** | 5.0 | **90.3** | 93.6 | 95.0 | **68.5** |

**Detection Performance.** Traditional methods such as LGTF and ADE achieve high recall ($\geq 96\%$) but suffer from extremely high false alarm rates (65–100%), leading to poor overall accuracy ($\leq 48.8\%$). Zero-shot LLMs (DeepSeek-R1, Qwen3-235B-A22B) provide more balanced results but remain unstable, with weak calibration and low overall accuracy ($< 10\%$). Instruction-tuned mid-scale LLMs reduce the false alarm rate (close to 0%) but significantly underfit long EDR sequences, resulting in binary accuracy around 60–69%. In contrast, WatchLog maintains both high recall and a low false alarm rate, achieving substantially better overall accuracy than all baselines.

**Reasoning Quality.** In addition to detection, we evaluate the generated rationales using three metrics: *Coherence*, *Consistency*, and *Completeness*. As reported in Table 2, WatchLog achieves the highest *Completeness* score (68.5), substantially outperforming all LLM baselines. Although competing models exhibit relatively strong *Coherence* and *Consistency* (e.g., Qwen2.5-1.5B and Qwen2.5-7B), they frequently miss key causal chains—the most essential aspect reflected by the *Completeness* metric. These results highlight that our implicit log-to-video representation provides stronger semantic grounding, enabling WatchLog not only to detect malicious activities but also to generate interpretable explanations crucial for real-world EDR applications.

Table 3: Performance on the **independent test set** and **unknown attack (OOD) test set**.

| Method | Independent Test Set | | | | Unknown Attack Test Set | |
|---|---|---|---|---|---|---|
| | FA↓ | R↑ | BA↑ | OA↑ | R↑ | Comp.↑ |
| Qwen2.5-1.5B | 24.0 | 73.0 | 72.7 | 45.5 | 32.0 | 39.6 |
| Qwen2.5-7B | **0.0** | 58.0 | 60.4 | 52.8 | 21.0 | 35.6 |
| LLaMA3.1-8B | 2.0 | 61.0 | 62.6 | 51.9 | 17.0 | 39.0 |
| **WatchLog-2B (Ours)** | 4.0 | **100.0** | **99.6** | **88.2** | **74.0** | **40.2** |

**OOD Generalization.** Due to the adversarial nature of cybersecurity, where attackers continuously modify existing malware or create entirely new variants to evade detection, evaluating on out-of-distribution (OOD) attacks is of critical importance for EDR systems. As shown in Table 3, while all baseline models achieve moderate performance on the independent test set (e.g., Qwen2.5-1.5B

reaches 73.0% recall but suffers from a high 24.0% false alarm rate, and LLaMA3.1-8B achieves balanced accuracy of 62.6%), their recall drops sharply on the OOD test set (falling to 21.0–32.0%). This indicates weak adaptability to novel threats. In contrast, WatchLog-2B maintains a near-perfect 100.0% recall and 99.6% balanced accuracy on the independent test set with only 4.0% false alarms, while also sustaining 74.0% recall and the highest *Completeness* score (40.2) on OOD attacks. These results demonstrate that WatchLog not only ensures reliable recognition under matched conditions but also generalizes effectively to previously unseen attacks, providing robustness and transferability that are indispensable for real-world deployment.

**Evaluation on External ATLASv2 Dataset.** To assess robustness beyond our curated dataset, we directly evaluate on the external **ATLASv2** benchmark in a zero-shot setting. Due to its limited size, no model was fine-tuned on this dataset. Results in Table 4 show distinct biases: Qwen2.5-1.5B achieves 100% recall by over-predicting malicious logs, while Qwen2.5-7B minimizes false alarms but misses most attacks. LLaMA3.1-8B offers a middle ground, yet overall accuracy remains low. Our **WatchLog** strikes the best balance, yielding 88% recall and

Table 4: Evaluation on ATLASv2. In our setting, we use only the Carbon Black Cloud subset of ATLASv2, which is a binary classification task (benign and malicious).

| Method | FA ↓ | R ↑ | BA ↑ |
|---|---|---|---|
| Qwen2.5-1.5B | 100.0 | **100.0** | 44.4 |
| Qwen2.5-7B | **5.0** | 13.0 | 58.3 |
| LLaMA3.1-8B | 85.0 | 75.0 | 41.7 |
| **WatchLog-2B (Ours)** | 50.0 | 88.0 | **66.7** |

the highest binary accuracy (66.7%), confirming strong generalization under distribution shift.

**Empirical Validation of Complexity.** Table 5 empirically verifies these trends by reporting GPU memory usage (GPU-MU) and latency (time-to-first-token, TTFT). We measure GPU memory consumption and first-token latency across varying input lengths (32K, 64K, 128K, 512K, 1024K). While baseline LLMs (e.g., Qwen2.5-7B, LLaMA3.1-8B) show near-linear or super-linear growth, our 2B model remains stable, with GPU-MU only increasing from 11.27 GB to 48.61 GB and TTFT from 0.63s to 1.11s. These results confirm that our compression (Stage 1) and temporal pooling with selective KV aggregation (Stage 2) effectively bound the sequence length $t \ll T$, allowing efficient reasoning on million-token logs.

Table 5: Comparison of GPU memory usage (GPU-MU, in GB) and inference latency (TTFT, in seconds) under different input lengths.

| | GPU-MU (GB) | | | | | TTFT (s) | | | | |
|---|---|---|---|---|---|---|---|---|---|---|
| | 64K | 128K | 256K | 512K | 1024K | 64K | 128K | 256K | 512K | 1024K |
| Qwen2.5-1.5B | 9.03 | 15.09 | 26.87 | 50.77 | 98.46 | 1.60 | 5.51 | 20.49 | 78.81 | 259.30 |
| Qwen2.5-7B | 27.13 | 39.67 | 64.74 | 114.88 | 215.17 | 4.33 | 13.99 | 50.23 | 151.76 | 486.19 |
| LLaMA3.1-8B | 31.02 | 46.69 | 78.03 | 140.71 | 266.07 | 5.31 | 17.61 | 53.73 | 167.43 | 521.72 |
| **WatchLog-2B (Ours)** | **11.27** | **12.67** | **15.47** | **26.81** | **48.61** | **0.63** | **0.66** | **0.69** | **0.79** | **1.11** |

## 4.3 ABLATION STUDY

**Different Alignment Pre-train Strategies.** We further analyze the contribution of each component in Watchlog through ablation experiments. First, we study the effect of alignment pre-train strategies (Table 6). We compare models trained without pre-training, with only image-level alignment, and with both image- and video-level alignment (default). Training without any alignment pre-train leads to severe degradation, as the model struggles to structure long, noisy logs. Introducing image–event alignment already leads to a substantial performance gain, and further adding video–log alignment yields the best results, suggesting that the two alignment stages are complementary.

Table 6: Ablation study on the impact of different alignment pre-train strategies.

| Method | FA ↓ | OA ↑ | Coh. ↑ | Cons. ↑ | Comp. ↑ |
|---|---|---|---|---|---|
| w/o alignment pre-train | 100.0 | 5.3 | 94.5 | 94.2 | 32.1 |
| + Image alignment pre-train | 5.0 | 88.8 | 92.1 | 93.6 | 67.2 |
| + Video alignment pre-train | 5.0 | 90.3 | 93.6 | 95.0 | 68.5 |

**Temporal Transformation Value $t$ (Figure 3 (a)).** In the first part of the ablation study, we focus on varying the temporal transformation value $t$, which controls the degree of temporal abstraction. As shown in the top row of Figure 3 (a), smaller values of $t$ imply a higher degree of temporal compression and lower computational cost, whereas larger values preserve more information but incur increased computational overhead. Notably, $t = 64$ strikes the best balance, as it provides a moderate level of temporal abstraction, preserving the most discriminative temporal cues while reducing the noise introduced by long event sequences. This demonstrates the critical role of the temporal transformation ratio in optimizing the performance of our framework. The exact numerical results underlying Figure 3 (a) are provided in Appendix Tables 13.

**Event Extension to Long Contexts (Figure 3 (b)).** In the second part of the ablation study, we evaluate the robustness of our framework to long-horizon contexts by extending event sequences by $2\times$, $3\times$, and $4\times$, simulating long contexts. As seen in the bottom row of Figure 3 (b), despite the substantial increase in sequence length, the performance degradation remains marginal. This demonstrates that our log-to-video representation and temporal aggregator are highly effective at preserving robustness and scalability under extreme temporal horizons. The results underline the framework's ability to handle long sequences without significant performance loss. The exact numerical results underlying Figure 3 (b) are provided in Appendix Tables 14.

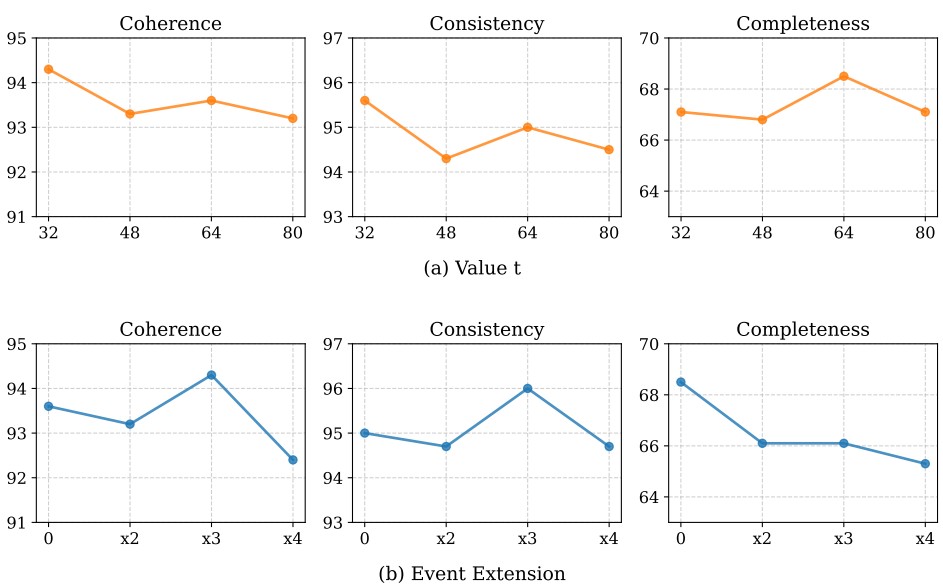

Figure 3: Ablation studies on temporal transformation ratio and event extension

The above ablation analyses clearly demonstrate the importance and effectiveness of each design choice in WatchLog. First, alignment pre-training is shown to be indispensable: removing it leads to substantial performance degradation, whereas incorporating both image-level and video-level alignment delivers the strongest results, confirming that the two stages provide complementary benefits for temporal semantic grounding. Second, the temporal transformation ratio $t$ plays a critical role in balancing abstraction and redundancy. Our results indicate that $t = 64$ achieves the best trade-off, preserving discriminative temporal cues while effectively filtering out noise and reducing computational overhead. Finally, by extending event sequences to test long-horizon contexts, we find that WatchLog sustains robust performance with only marginal degradation, demonstrating strong scalability and resilience even under extreme temporal lengths. Together, these findings validate the effectiveness of our design and highlight its robustness across a wide range of operating conditions.

**VLM Backbone Scaling**. To assess the impact of the underlying VLM backbone on WatchLog, we replace the default model with increasingly stronger variants, including Qwen2.5-VL-3B-Instruct and Qwen2.5-VL-7B-Instruct Yang et al. (2024), and conduct evaluations under identical settings. As shown in Table 7, scaling up the backbone leads to consistent improvements across all key metrics. Moving from 2B to 3B eliminates false alarms entirely (5.0 vs 0.0) and brings notable gains in overall accuracy (90.3 vs 91.3), *Coherence* (93.6 vs 95.8), *Consistency* (95.0 vs 96.6), and

*Completeness* (68.5 vs 74.4). Further scaling to the 7B backbone brings additional gains, achieving the highest detection overall accuracy of 92.5% and reasoning quality scores of 96.3, 97.3, and 78.2 in *Coherence*, *Consistency*, and *Completeness*, respectively.

These results demonstrate that WatchLog not only generalizes effectively to larger VLMs but also benefits substantially from increased model capacity, with gains that are both stable and significant, particularly in the accuracy of generated reasoning rationales. Notably, these improvements show that WatchLog can be easily and effectively adapted to stronger foundation models without requiring any architectural changes.

Table 7: Ablation study on VLM backbone scaling.

| Method | FA ↓ | OA ↑ | Coh. ↑ | Cons. ↑ | Comp. ↑ |
|---|---|---|---|---|---|
| WatchLog-2B | 5.0 | 90.3 | 93.6 | 95.0 | 68.5 |
| WatchLog-3B | 0.0 | 91.3 | 95.8 | 96.6 | 74.4 |
| WatchLog-7B | 0.0 | 92.5 | 96.3 | 97.3 | 78.2 |

**Spatial Arrangement Sensitivity**. To investigate the robustness of our method to spatial arrangement sensitivity, we perform an ablation study with two experimental settings involving randomly permuted patch layouts. In the first setting, the model is trained with randomly permuted patches but tested using the default layout (alphabetically by key). In the second setting, both training and testing involve randomly permuted patches, and we report the averaged results over five runs.

As shown in Table 8, the results demonstrate that patch permutation substantially reduces the false alarm rate, albeit with a slight decrease in overall accuracy. For the reasoning-quality metrics, applying random permutation only during training improves *Coherence* and *Consistency*, with a modest decrease in *Completeness*. When permutation is applied during both training and testing, the reductions in reasoning-quality metrics remain within acceptable bounds.

Overall, these results demonstrate that WatchLog is robust to spatial arrangement variations, maintaining high performance even when patch layouts are randomized. The minor fluctuations observed further highlight the flexibility and resilience of our approach, making it effective even when the spatial arrangement of input patches changes.

Table 8: Ablation study on spatial arrangement sensitivity.

| Method | FA ↓ | OA ↑ | Coh. ↑ | Cons. ↑ | Comp. ↑ |
|---|---|---|---|---|---|
| WatchLog-2B (default) | 5.0 | 90.3 | 93.6 | 95.0 | 68.5 |
| WatchLog-2B (perm. train only) | 0.0 | 88.5 | 94.2 | 96.2 | 67.1 |
| WatchLog-2B (perm. train & test) | 1.0 | 88.0 | 92.9 | 94.6 | 64.8 |

## 5 CONCLUSION

In this paper, we presents **WatchLog**, a novel framework that reformulates endpoint security logs into video-language representations for malicious behavior detection. WatchLog consistently outperforms both baseline models and LLM-based approaches across multiple performance metrics, including binary accuracy, recall, and overall accuracy. Extensive experiments further demonstrate that WatchLog delivers strong detection accuracy and high-quality reasoning, producing results that are both robust and interpretable. Moreover, our real-world testing in an enterprise endpoint detection environment validates WatchLog's efficiency, effectiveness, and strong generalization ability in identifying both known and unknown attacks. Looking forward, we plan to enhance WatchLog by incorporating deeper cybersecurity domain knowledge to further strengthen its reasoning capabilities, and by exploring its integration with broader security solutions to support more comprehensive and adaptive defense systems.

## ETHICS STATEMENT

All data used in this study were collected and processed in compliance with relevant privacy and security regulations. No personally identifiable information is included in the datasets, and all experiments were conducted on anonymized system logs. Our method is intended for research in cybersecurity and endpoint detection, and we emphasize responsible use. In preparing this manuscript, we made very light use of a large language model solely to polish a few sentences for clarity; all scientific content, methods, and experimental results were independently developed by the authors.

## REPRODUCIBILITY STATEMENT

We provide detailed descriptions of our datasets, preprocessing steps, and feature extraction in Section 4.1 and Appendix D. Model architectures, training procedures, hyperparameters, and supervised fine-tuning schedules are fully specified in Section 4.1. Mathematical properties, complexity analysis, and stability guarantees of each stage are provided in Appendix H to support the theoretical understanding of our approach. The full implementation and training scripts will be made publicly available to facilitate reproduction of all reported results.

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

## A  USE OF LARGE LANGUAGE MODELS IN PAPER PREPARATION

We made very limited use of a large language model (LLM) to assist with minor sentence-level edits for clarity and readability. The LLM was only used to improve grammar and expression; all scientific content, technical methods, experiments, and analyses were entirely developed by the authors. The final manuscript was carefully reviewed by the authors to ensure correctness and fidelity to the original work. No LLM was used to generate novel technical content, mathematical derivations, or experimental results.

## B  PROMPT FOR GENERATING EVENT DESCRIPTIONS

Table 9: Prompts used for event descriptions generation.

---

You are a professional cybersecurity analyst. Given the following EDR log (a single event), generate a professional text description.

###Requirements:
1. Base the description only on the explicit information in the log. Do not invent or speculate.
2. Use clear, concise, full sentences. Avoid redundancy.
3. Cover all key elements present in the log, such as timestamp, host, process, command line, file activity, network connections, and hash values.
4. Output should be a single natural-language paragraph, not a list or JSON.

### Input
EDR log event: [*event*]

---

## C  PROMPT FOR GENERATING LOG STORIES

Table 10: Prompts used for log stories generation.

---

You are a professional cybersecurity analyst. You are given a log consisting of multiple paragraphs, where each paragraph represents a text report recorded at different time points. Your task is to generate a comprehensive yet concise summary of the entire log.

###Requirements:
1. Completeness: Cover all major points and important details from the entire log without omitting key information.
2. Conciseness: Ensure the summary remains clear, well-structured, and free of redundancy, while still preserving important details.
3. Coherence: Present the summary in a logically organized and fluent manner, not as a fragmented list of sentences.
4. Fidelity: Do not introduce assumptions, interpretations, or information that is not present in the original log.

### Input
LOG: [*log*]

---

## D  DATA COLLECTION

The availability of large-scale and high-quality datasets, particularly those encompassing diverse attack types and complex behavioral patterns, is essential for validating algorithms and enhancing the detection capabilities of EDR systems. To address the limitations of existing public EDR datasets in terms of scale, diversity, and realism, we developed a custom tool for real-time monitoring and capture of system activities.

Specifically, we used virtualization technology to build multiple realistic enterprise environments, ensuring both the safety and authenticity of data collection. Automated scripts were used to execute well-defined attack scenarios, including malware injection, privilege escalation, and lateral movement, enabling the capture of complete execution traces of real-world malicious behaviors. Raw data was collected via the Windows kernel callback mechanism and stored in JSON format in real time, ensuring comprehensive and fine-grained behavioral records. To maintain data integrity and support real-time processing, we utilized a high-efficiency non-paged memory pool for event caching and designed a circular buffer to optimize data transmission.

All samples and their corresponding behavior chains were meticulously labeled by security experts to guarantee annotation accuracy. The resulting dataset contains over 8 million event records, covering both normal activities and 172 distinct real-world malicious families. For our experiments, we select the 20 most frequent behavior families (including normal activities) to construct standard training and testing sets, while the remaining 153 families are treated as unknown attacks. Compared to existing public EDR datasets, EDR8M-20R offers a distinctive combination of scale, event completeness, behavioral chain granularity, and annotation quality, making it one of the most comprehensive publicly available datasets of its kind.

## E  BEHAVIORAL RATIONALE CONSTRUCTION

To enhance the interpretability of EDR detection results during the SFT stage, we construct behavioral rationale data that explicitly explain why a given log is associated with a specific attack or activity family. Specifically, for each log, we design a structured prompt to guide GPT-5 in generating these rationales. Each prompt consists of three components: (1) descriptions of the individual events, (2) the target behavioral family, and (3) a brief description of the family's characteristics. The prompt is formulated as follows:

Table 11: Prompts used for rationale data generation.

| **Prompt for a Specified Behavioral Family** |
| --- |
| You are a professional cybersecurity analyst. You are given a log consisting of multiple paragraphs, where each paragraph is a text report recorded at different time points. 

 You are also provided with: 
 -A behavior family label. 
 -A brief description of this family. 

 Explain briefly and directly why the log belongs to the behavioral family, focusing only on concrete evidence from the log that matches the description. Write the explanation in a concise and professional style. 

 ### Input 
 LOG: [*log*] 
 Behavioral Family: [*family name*] 
 Description of [*family name*]: [*family description*] |
| **Prompt for Unknown Attack Family** |
| You are a professional cybersecurity analyst. You are given a log consisting of multiple paragraphs, each recorded at different time points. 

 The log contains suspicious activity, but the type of attack is unknown. Analyze the log and provide a concise, professional explanation of why this log indicates an attack, focusing only on concrete evidence from the log itself. 

 ### Input 
 LOG: [*log*] |

## F    IMPLEMENTATION DETAILS OF THE BASELINE METHODS

In this work, we compare several state-of-the-art machine learning and deep learning methods in computer security, as well as recent approaches based on large language models (LLMs). The implementation details of each method are described below.

**LGTF** Kumar et al. (2022) is a malware detection method that leverages texture features extracted from images for effective pattern classification across multiple benchmark datasets. To adapt LGTF to our EDR dataset, we uniformly sampled 128 events per sample and transformed the logs into image sequences. Each character in an event is encoded in UTF-8 and mapped to a pixel in a predefined $56 \times 56$ image, with pixels arranged according to a fixed key order. Following the optimal configuration reported in the original work, we extract global features from each image, concatenate them, and apply the Bag-of-Visual-Words (BoVW) algorithm for dimensionality reduction. Finally, a Support Vector Machine (SVM) classifier is trained with tuned hyperparameters to achieve optimal performance.

**ADE** is a generalizable malicious URL detection method proposed by Tsai et al. (2024). In our experiments, we adopted the MalConv baseline model Catanzaro & Nicholas (2018) following the implementation details provided in Tsai et al. (2024), and applied the Adversarial Debiasing Embedding (ADE) training strategy for performance evaluation. All other hyperparameters remain consistent with the original configuration.

**DrSec** Sharif et al. (2024) is a malicious behavior detection method based on pre-trained language models. Following the original implementation, we adopted the RoBERTa architecture to extract features from log text. To address the input context length limitation of RoBERTa, a sliding window strategy is applied for sequence segmentation. A shallow neural network classifier is attached to the output layer for end-to-end fine-tuning. All hyperparameters are kept consistent with the original settings, and we report the best evaluation results.

**DeepSeek-R1** Guo et al. (2025) is a general-purpose reasoning model that achieves strong performance across various domains, surpassing OpenAI o1 Jaech et al. (2024) on several benchmark tasks. Due to its substantial computational demands, we evaluated the full 671B-parameter model in a zero-shot setting. The input context length is set to the model's maximum of 128K tokens, with a temperature of 0.6 and a top-$p$ of 0.95. To constrain predictions to a predefined set of behavioral categories, all candidate family names are appended to the input prompt.

**Qwen3-235B-A22B** Yang et al. (2025) is a large language model developed by the Qwen team, offering dense and mixture-of-experts (MoE) architectures. Built upon extensive training, it demonstrates substantial improvements in reasoning, instruction-following, and multilingual support. We also evaluated Qwen3-235B-A22B in a zero-shot setting, using the same hyperparameter configurations and prompts as in the evaluation of DeepSeek-R1.

**Qwen2.5-1.5B/7B** Yang et al. (2024) is an open-source general-purpose language model. Through supervised fine-tuning, it has been successfully applied to various domain-specific tasks. In our experiments, we fine-tune the model using the same training set and perform inference with a 128K token context length. All other hyperparameters are kept consistent with those described in the experimental setup to ensure comparability and reproducibility.

**LLaMA3.1-8B** Grattafiori et al. (2024) is an open-source general-purpose language model developed by Meta AI, showing strong performance and generalization in various NLP tasks. We fine-tuned LLaMA3.1-8B on the same training data, using parameter configurations identical to those used for Qwen2.5-1.5B/7B to ensure fair comparison and reproducibility. We also perform inference at a 128K token context length.

## G    PROMPT FOR RATIONALE EVALUATION

In our study, we evaluate the quality of generated rationales along three dimensions: **Coherence**, **Consistency**, and **Completeness**. This evaluation is particularly critical in the context of malicious behavior detection, where end users (e.g., security analysts) must not only trust the final predictions but also rely on interpretable reasoning processes to validate or further investigate the system's decisions.

Specifically, *Coherence* measures the logical flow and internal readability of a rationale, *Consistency* assesses whether the rationale faithfully supports the predicted label, and *Completeness* captures the degree to which the rationale aligns with a reference rationale.

To ensure reproducibility and reduce subjective bias, we employ GPT-5 as an automatic evaluator. Table 12 presents the prompts used for the evaluation. Each prompt instructs the evaluator to provide both a concise textual justification (2–3 sentences) and a numerical rating on a 1–5 Likert scale.

# H  THEORETICAL AND MATHEMATICAL ANALYSIS OF EACH STAGE

## H.1  STAGE 1: EVENT-TO-IMAGE VIA KV2EMBEDDING TRANSFORMATION

Given an event $E$ decomposed into key–value pairs $\{\mathbf{x}_m = (k_m, v_m)\}_{m=1}^M$ and token embeddings $e_{m\ell} \in \mathbb{R}^d$, the kvEmbedding layer produces

$$\tilde{e}_m \;=\; \sum_{\ell=1}^{L_m} \alpha_{m\ell}\, e_{m\ell}, \qquad \alpha_{m\ell} \;=\; \frac{\exp(W^\top e_{m\ell})}{\sum_{u=1}^{L_m} \exp(W^\top e_{mu})},$$

with $W \in \mathbb{R}^d$. Two elementary but important properties follow immediately from this definition. First, $\tilde{e}_m$ is a convex combination of the tokens $\{e_{m\ell}\}$; consequently $\tilde{e}_m$ lies in the convex hull of the token vectors and in particular its norm is bounded by the largest token norm, i.e. $\|\tilde{e}_m\| \leq \max_\ell \|e_{m\ell}\|$. Second, the mapping from the multiset $\{e_{m\ell}\}_{\ell=1}^{L_m}$ to $\tilde{e}_m$ is permutation-invariant with respect to the token ordering: $\tilde{e}_m$ depends only on the multiset of token vectors and not on their sequence index, because both the logits $W^\top e_{m\ell}$ and the convex combination are point-wise functions of each token. These properties imply that kvEmbedding provides a stable, order-agnostic pooling of variable-length values into a fixed-dimensional representation; stability here is a direct consequence of the softmax normalization (bounded coefficients) and differentiability.

From a computational viewpoint, computing $\{\tilde{e}_m\}_{m=1}^M$ requires a linear pass over the tokens: each token needs the dot product $W^\top e_{m\ell}$ and a weighted sum, so the per-event cost scales as $O\big(\sum_{m=1}^M L_m \cdot d\big)$. After projection into the ViT space and reshaping the $M$ vectors into $\tilde{E} \in \mathbb{R}^{c \times h \times w}$ (with $M = hw$), the visual footprint of a single event is $O(c \cdot h \cdot w)$ in activation memory.

The image–description objective is an autoregressive negative log-likelihood

$$\mathcal{J}_{IEA} \;=\; -\sum_{l=1}^{L_I} \log P_{\theta'}(S_l \mid S_{<l}, E).$$

Viewed probabilistically, minimizing the expected value of $\mathcal{J}_{IEA}$ over the data distribution is equivalent to minimizing the conditional Kullback–Leibler divergence between the empirical conditional distribution of descriptions and the model conditional distribution induced by $\tilde{E}$. Formally,

$$\mathbb{E}_{(E,S)\sim\mathcal{D}}\big[\mathcal{J}_{IEA}\big] = \mathbb{E}_E\big[H\big(p_{\text{data}}(\cdot \mid E)\big)\big] + \mathbb{E}_E\big[D_{\text{KL}}\big(p_{\text{data}}(\cdot \mid E) \,\big\|\, p_{\theta'}(\cdot \mid \tilde{E})\big)\big],$$

where $p_{\text{data}}(S \mid E)$ denotes the (possibly stochastic) distribution of natural-language descriptions for event $E$. Thus training with $\mathcal{J}_{IEA}$ explicitly encourages the learned visual encoding $\tilde{E}$ to be predictive of human-interpretable descriptions in the sense of minimizing the conditional KL divergence. Practically, this provides a semantically rich inductive bias: the encoder must produce features that preserve the aspects of the event that are informative for language generation. Because the kvEmbedding is permutation-invariant and norm-bounded, it yields numerically stable gradients and avoids exploding activations in this alignment step.

This stage therefore constructs compact, semantically grounded event images $\tilde{E}$ with controlled computational cost and a clear maximum-likelihood interpretation; these images serve as the atomic visual primitives that the temporal stage will aggregate.

## H.2  STAGE 2: LOG-SEQ → VIDEO VIA TEMPORAL CROSS-ATTENTION

Stacking the per-event images produces a video-like tensor $\mathcal{E} = [\tilde{E}_1, \ldots, \tilde{E}_T] \in \mathbb{R}^{T \times c \times h \times w}$. To reduce temporal redundancy we form a small set of queries $Q = \mathcal{E}_{\text{pool}} \in \mathbb{R}^{t \times c \times h \times w}$ (with $t \ll T$)

Table 12: Prompts used for rationale evaluation in terms of Coherence, Consistency, and Completeness.

| Metric | Prompt Description |
| --- | --- |
| Coherence | You are an expert evaluator for large language model reasoning. Your task is to assess the **coherence** of the given rationale.
Coherence means that the rationale should be internally logical, step-by-step, and free from contradictions or abrupt jumps.

### Evaluation Criteria
1. Highly coherent: The rationale is well-structured and flows logically.
2. Moderately coherent: The rationale is mostly logical but has minor gaps, redundancies, or unclear transitions.
3. Poor coherence: The rationale is disorganized, contradictory, or difficult to follow.

### Input
- Model's Rationale: [*rationale*]

### Output Format
Provide:
- A short explanation of your judgment (2-3 sentences).
- A rating on a 1-5 scale (1 = very poor coherence, 5 = excellent coherence). |
| Consistency | You are an expert evaluator for large language model reasoning. Your task is to assess the **consistency** between the given rationale and the predicted answer.
Consistency means that the rationale should logically support the answer, without contradictions.

### Evaluation Criteria
1. Fully consistent: The rationale clearly and logically leads to the answer.
2. Partially consistent: The rationale provides partial support for the answer, but has gaps or irrelevant reasoning.
3. Inconsistent: The rationale contradicts the answer or fails to support it logically.

### Input
- Model's Answer: [*answer*]
- Model's Rationale: [*rationale*]

### Output Format
Provide:
- A brief explanation of your judgment (2-3 sentences).
- A rating on a 1-5 scale (1 = fully inconsistent, 5 = fully consistent). |
| Completeness | You are an expert evaluator for large language model reasoning. Your task is to assess the **completeness and alignment** of the model-generated rationale compared to a reference rationale.
Completeness means that the model's rationale should include the key reasoning steps from the reference rationale, without omitting essential parts.

### Evaluation Criteria
1. Highly aligned: The model rationale includes all key reasoning steps and closely matches the reference rationale.
2. Partially aligned: The model rationale includes some but not all key reasoning steps.
3. Misaligned: The model rationale misses most key reasoning steps or introduces unrelated/incorrect reasoning.

### Input
- Reference Rationale: [*reference_rationale*]
- Model's Rationale: [*predict_rationale*]

### Output Format
Provide:
- A short explanation of your judgment (2-3 sentences).
- A rating on a 1-5 scale (1 = very poor alignment, 5 = highly aligned). |

and apply cross-attention at every spatial location:

$$\tilde{\mathcal{E}}_{i,j} = \text{Softmax}\left(\frac{Q_{i,j}W_Q(K_{i,j}W_K)^\top}{\sqrt{c}}\right)(V_{i,j}W_V), \qquad K, V = \mathcal{E},$$

so that for each pixel coordinate $(i, j)$ the condensed vectors $\tilde{\mathcal{E}}_{i,j}$ are linear combinations of the original temporal vectors $\{V_{i,j}(\tau)\}_{\tau=1}^T$. Consequently, at every spatial location the condensed representation lies in the linear span of the $T$ original temporal vectors; in matrix terms the cross-attention realizes a rank-at-most-$t$ projection of the $T$-frame sequence at each pixel. The rank bound implies a monotone relationship between the transformed size $t$ and representational capacity: increasing $t$ enlarges the span that can be represented and thereby reduces the approximation error (measured, e.g., in squared reconstruction error) of temporal dynamics that can be captured at that pixel. Conversely, choosing $t \ll T$ provides a controlled low-rank approximation that removes redundancy and focuses capacity on salient temporal patterns.

The computational cost of forming $\tilde{\mathcal{E}}$ dominates along three dimensions: the number of frames $T$, the condensed frame count $t$, and the spatial and channel dimensions $h, w, c$. A direct implementation computing attention scores for every $(i, j)$ incurs cost proportional to $O(h \cdot w \cdot c \cdot t \cdot T)$ for the main score multiplications, and requires activation storage roughly $O(T \cdot c \cdot h \cdot w)$ for keys and values. Using $t \ll T$ therefore translates into substantial arithmetic and memory savings in downstream processing, while the pixelwise attention ensures that spatial consistency is preserved (temporal aggregation is local in the spatial index and thus does not mix unrelated pixel locations).

The video–log objective is again an autoregressive NLL

$$\mathcal{J}_{VLA} = -\sum_{l=1}^{L_V} \log P_{\theta''}(\mathcal{S}_l \mid \mathcal{S}_{<l}, \mathcal{L}),$$

and the same KL-decomposition as in Stage 1 holds: minimizing $\mathcal{J}_{VLA}$ pushes the model conditional $p_{\theta''}(\cdot \mid \tilde{\mathcal{E}})$ toward the empirical conditional distribution of log narratives. Because $\tilde{\mathcal{E}}$ is formed from event-level encodings that were themselves grounded in language in Stage 1, the cross-attention step has the dual effect of (i) transforming temporal information into a compact video representation and (ii) preserving the semantic axes that are relevant for language generation. In particular, temporal cross-attention is time-aware (the key positions carry time indices) so the model can learn order-sensitive dynamics rather than a mere bag-of-frames summary.

Practically, this stage thus produces a temporally condensed but semantically aligned video tensor $\tilde{\mathcal{E}}$ that balances representational fidelity (controlled by $t$) against computational footprint, and it prepares a temporally coherent input for the final supervised stage.

### H.3 Stage 3: Supervised fine-tuning

In fine-tuning we train the multimodal model end-to-end to produce a family label $y$ and a natural-language rationale $r$. The joint autoregressive objective is

$$\mathcal{J}_{SFT} = -\sum_{l=1}^{L} \log P_{\theta^*}\big((y, r)_l \mid (y, r)_{<l}, (\mathcal{L}, \mathbf{q})\big).$$

Using the chain rule for probabilities the joint target factorizes as $P(y, r \mid \tilde{\mathcal{E}}, \mathbf{q}) = P(y \mid \tilde{\mathcal{E}}, \mathbf{q}) \cdot P(r \mid y, \tilde{\mathcal{E}}, \mathbf{q})$. Minimizing $\mathcal{J}_{SFT}$ therefore simultaneously improves the discriminative component $P(y \mid \cdot)$ and the conditional explanation model $P(r \mid y, \cdot)$. From a learning-theory perspective this is a form of multi-task or auxiliary-task training in which rationales act as a task-consistent regularizer: the need to generate coherent, label-consistent explanations constrains the function class and can reduce overfitting on the classification objective alone. In probabilistic terms, the expected supervised loss decomposes into the target entropy plus a KL term that measures the discrepancy between the empirical conditional distribution of $(y, r)$ and the model distribution; minimizing it yields a maximum-likelihood estimate for the joint target under the model family.

Gradient signals from $\mathcal{J}_{SFT}$ flow into both the language model parameters and the visual encoder that produced $\tilde{\mathcal{E}}$, thereby refining visual features to be discriminative for the final task while maintaining semantic alignment learned during pretraining. Because Stage 1 and Stage 2 already shaped

$\tilde{\mathcal{E}}$ to encode human-interpretable structure, the supervised optimization typically requires fewer task-specific examples to reach a given error level compared to training from random initialization: the pretraining stages provide an informative initialization (a form of data-dependent prior) that reduces effective sample complexity in standard transfer-learning regimes.

On convergence and practical behavior: each objective $\mathcal{J}_{IEA}, \mathcal{J}_{VLA}, \mathcal{J}_{SFT}$ is a sum of negative log-probabilities and thus is amenable to stochastic gradient optimization; while nonconvexity precludes global optimality guarantees, under common smoothness and bounded-variance assumptions gradient methods converge to stationary points. In practice, a stagewise schedule (first Stage 1, then Stage 2, finally Stage 3 fine-tuning) is well matched to the probabilistic decomposition above because it first enforces local semantic alignment (events), then temporal coherence (videos), and finally task discrimination and explanation.

# I SUPPLEMENTARY NUMERICAL RESULTS

To complement the visualizations in the main text, we provide the corresponding numerical values in tabular form. These tables ensure reproducibility and enable precise inspection of our results. Each table corresponds to one figure in the main body, and provides the exact scores underlying the plots.

## ABLATION ON TEMPORAL TRANSFORMATION LENGTH (FIGURE 3, TOP ROW)

Table 13 provides the raw numbers for the ablation study on temporal transformation length $t$, corresponding to the top row of Figure 3. We vary $t \in \{32, 48, 64, 80\}$ and report both detection metrics (False Alarm, Overall Accuracy) and reasoning scores (all rescaled to 0–100). The visualization in the main text illustrates overall trends, whereas this table allows precise comparison across different $t$ settings.

Table 13: Ablation study on the effect of temporal transformation length $t$ (corresponding to Figure 3, top row). Detection metrics are in percentage (%), and reasoning metrics are rescaled to 0–100.

| $t$ Value | FA $\downarrow$ | OA $\uparrow$ | Coh. $\uparrow$ | Cons. $\uparrow$ | Comp. $\uparrow$ |
|---|---|---|---|---|---|
| 32 | 10 | 87.5 | 94.0 | 95.2 | 66.7 |
| 48 | 5 | 89.5 | 92.8 | 94.1 | 66.6 |
| 64 | 5 | 90.3 | 93.2 | 95.0 | 68.1 |
| 80 | 5 | 89.8 | 93.1 | 94.2 | 66.7 |

## ABLATION ON SCALING FACTOR (FIGURE 3, BOTTOM ROW)

Table 14 corresponds to the bottom row of Figure 3, which studies the effect of scaling event-level cross-attention modules by different factors. We report detection and reasoning metrics for the default ($\times 1$) and scaled variants ($\times 2, \times 3, \times 4$). The table complements the main-figure visualization with exact values, making clear the slight performance trade-offs introduced by scaling.

Table 14: Ablation study on the effect of scaling the event-level cross-attention modules (corresponding to Figure 3, bottom row). Detection metrics are in percentage (%), and reasoning metrics are rescaled to 0–100.

| Scaling Factor | FA $\downarrow$ | OA $\uparrow$ | Coh. $\uparrow$ | Cons. $\uparrow$ | Comp. $\uparrow$ |
|---|---|---|---|---|---|
| 1 | 5 | 90.3 | 93.2 | 95.0 | 68.1 |
| $\times 2$ | 5 | 88.5 | 92.9 | 94.6 | 65.8 |
| $\times 3$ | 5 | 88.5 | 94.1 | 95.8 | 65.8 |
| $\times 4$ | 5 | 87.3 | 92.2 | 94.5 | 64.9 |

## J  VISUALIZATION OF LOG ANALYSIS

To provide an intuitive illustration of how our framework interprets malicious behaviors, we present a visualization of log analysis in Table 15 to Table 17. For clarity and readability, certain field values (e.g., MD5 values and file paths) are partially masked with "..." placeholders. This visualization shows that our approach not only achieves accurate detection, but also delivers interpretable evidence of how malicious behaviors evolve over time.

Table 15: Visualization of our model's reasoning on a log sample from the *Win32_Worm.FakeFolder* attack. Text highlights with the same color indicate identical rationale semantics.

---

**Input Log: [*An ultra-long log sequence...*]**

---

Ground-Truth family:   Win32_Worm.FakeFolder
Ground-Truth rationale:
  -Self-replication/propagation:   On 2024-04-17, a PE file named 9d82...e9e2.exe is written to D:\sample\9d82....e9e2.exe, indicating copying a main payload to another location for propagation.
–Persistence via Run keys:   Recurrent AsepValueUpdate events show the registry Run key HKEY_LOCAL_MACHINE\...\Windows\CurrentVersion\Run updated to load system.exe (e.g., C:\Windows\system32\system.exe or C:\Windows\syswow64\system.exe) across multiple dates (May–June 2024). This demonstrates startup persistence typical of FakeFolder behavior.
  -DLL-based execution chain: Multiple LoadImage and AsepValueUpdate entries show rundll32.exe executing C:\Windows\system32\rlfmb.dll Exucute and loading other DLLs (e.g., ifaja.dll, rlfmb.dll). This indicates payload execution via DLLs loaded through Rundll32, a common technique for FakeFolder variants.
  -Deceptive/masquerading and complex process chain: The event sequences consistently show long, multi-step process chains (smss.exe → winlogon.exe → userinit.exe → explorer.exe, etc.) preceding the malicious actions, consistent with a worm attempting to blend into system startup and user sessions.
  In sum, the logs show self-replication attempts, repeated Run-key persistence, and DLL-based execution paths, all of which map to the described behavior of Win32_Worm.FakeFolder.

---

Our predictions:   Win32_Worm.FakeFolder
Our rationale:
  -Self-replication/propagation indicators: The same binary (7f...9f.exe) is repeatedly created and executed from D\:\sample\with different timestamps, suggesting replication to local drives rather than just memory. The same MD5 hash is used across events, confirming a single binary propagating across the host.
  -Masquerading and persistence via startup mechanisms: Multiple AsepValueUpdate events show the registry Run key (HKEY_LOCAL_MACHINE\...\Microsoft\Windows\CurrentVersion\Run) being set to launch C:\Windows\system32\system.exe, i.e., a persistence mechanism designed to auto-start on logon. This is a classic FakeFolder tactic to ensure re-execution after reboot.
  -Use of rundll32 to execute payloads: Repeated LoadImage events show rundll32.exe loading and executing wgbhf.dll (Exucute), with command lines like C:\Windows\system32\rundll32.exe Exucute (and variants). This DLL loading pattern is typical for payload deployment executed via legitimate system processes.
  -Process chain and stealth/persistence pattern: The events describe a long process chain (smss.exe → winlogon.exe → userinit.exe → explorer.exe → system.exe → rundll32.exe, etc.) with high integrity levels and extensive cleanup-write actions, consistent with attempts to establish persistence and concealment.
  In summary, the log shows self-replication to local drives, Run-key persistence for auto-start, and rundll32-based payload execution, all characteristic of Win32_Worm.FakeFolder.

---

Table 16: Visualization of our model's reasoning on a log sample from the *Win32_TrojanDownloader.Upatre* attack. Text highlights with the same color indicate identical rationale semantics.

---

**Input Log: [*An ultra-long log sequence. . .* ]**

---

Ground-Truth family: Win32_TrojanDownloader.Upatre

Ground-Truth rationale:

-Masquerading dropper: Numerous entries depict executables named WinWord.exe (Office 2007) and related temporary variants (e.g., 86EC1D. . . , DE77.tmp, DC92.tmp, 4F71.tmp, etc.) being created and executed from the user/admin Temp paths. This pattern fits Upatre's practice of dropping a loader under a trusted-looking name to bypass user suspicion.

-On-disk payload writes in Temp: Repeated PE write events (PeFileWritten) with FILE_OP_FLAG_WRITTEN_ON_CLEANUP in C:\Users\admin\AppData\Local\Temp and similar Temp folders indicate multiple payloads being written during cleanup phases, consistent with a downloader staging subsequent stages.

-Multi-stage process chains: The log shows long chains of temporaries and chained processes (e.g., A31E.tmp → A69A.tmp → 3BD4.tmp → 2D4D.tmp, etc.), a hallmark of downloader activity that drops and executes successive payloads.

-Elevated/hidden execution context: Many processes run with high integrity and network level 70, and several are launched without command lines, suggesting stealthy stages and potential C2/download activity.

In short, the observations match Upatre's typical behavior pattern: initial access via a disguised dropper, on-disk multi-stage payload delivery in Temp, extensive use of chained temporary executables, and implied network-based payload retrieval, all within a Word/Office context.

---

Our predictions: Win32_TrojanDownloader.Upatre

Our rationale:

-On-disk dropper activity in Temp with obfuscated filenames and repeated PE writes (PeFileWritten) in a long process chain. -The log shows numerous temporary PE files being created and written in the user Temp folder (for example: 9A1.tmp, 3D21.tmp, 62B3.tmp, 7A1.tmp, etc.), each associated with WinWord.exe from the 2007 Office suite. These files are written to C:\Users\admin\AppData\Local\Temp\ and then later written to disk, consistent with Upatre's behavior of dropping multiple staged PE payloads on disk.

-Multi-stage, long process chains centered on WinWord.exe from Office 2007.

-A long chain of temporaries (e.g., 9A1.tmp → 3D21.tmp → 62B3.tmp → 7A1.tmp → 1E08.tmp → 22B.tmp → 3034.tmp → 337.tmp → 3691.tmp → 39E.tmp → 42B.tmp → 449.tmp, etc.) all referencing WinWord.exe from the 2007 Office suite. This pattern matches Upatre's multi-stage downloader that fetches and executes a second-stage payload by abusing a chain of temporary processes.

-High integrity, network-capable processes forming the downloader chain.

- The involved processes in the chain (WinWord.exe) appear with high integrity and a network level of 70 across many events, consistent with Upatre's behavior of network-enabled download/drop operations and subsequent payload execution.

- Cleanup-time PE writes (FILE_OP_FLAG_WRITTEN_ON_CLEANUP) indicative of staged payloads.

- Numerous PeFileWritten events show PE files being written during cleanup (for example: 9A1.tmp writes 9D07.tmp; 3D21.tmp writes 3E11.tmp; and so on). This aligns with Upatre's technique of dropping and finalizing second-stage payloads as cleanup occurs.

In short, the log exhibits Upatre-like behavior: extensive on-disk, multi-stage PE drops in Temp, a long chain of Office-temporary processes referencing WinWord.exe, and cleanup-time file writes that together reflect the downloader-and-payload-fetch pattern of Win32_TrojanDownloader.Upatre.

Table 17: Visualization of our model's reasoning on a log sample from the *Worm.Win32.Scar* attack. Text highlights with the same color indicate identical rationale semantics.

---

**Input Log: [An ultra-long log sequence. . . ]**

---

Ground-Truth family:  Worm.Win32.Scar
Ground-Truth rationale:
  -Self-propagation/replication signals in a single host:
  - Repeated creation and execution of numerous Sysqem*.exe binaries located in C:\Users\admin\AppData \Local\Temp\ across many timestamps, each with high integrity and network level 70.  This pattern is consistent with Scar's self-replicating payloads and stagedDroppers that proliferate multiple components on a host.
  -File-based propagation with cleanup writes (typical Scar tactic):
  - Frequent PE file writes in Temp directories labeled as PeFileWritten with FILE_OP_FLAG_WRITTEN_ON _CLEANUP, often accompanied by long chains of Sysqemxxx processes.
    This indicates dropping and cleaning up multiple executable components rather than relying on registry changes or explicit network actions.
  -Complex process chains and persistence-oriented execution flow:
  Logs show extended chains from system processes (smss.exe, winlogon.exe, explorer.exe) through multiple Sysqem-related executables,  suggesting a staged, multi-component infection flow typical of Scar's approach to persistence and payload deployment.
  -Lack of explicit user-driven commands or targeted network actions:
    While network level values exist, the events emphasize file drops and process chaining over command-line payloads or registry edits, aligning with Scar's basic self-propagation behavior rather than a targeted, user-initiated intrusion.
  In sum, the behavior—large-scale, internal propagation via many Sysqem*.exe Dropper/Payload components in Temp with cleanup-file writes, and long process chains originating from core Windows processes—matches the Scar worm's core traits (self-propagation, self-replication, staged drops, lightweight persistence) as described for Worm.Win32.Scar.

---

Our predictions:  Worm.Win32.Scar
Our rationale:
  - The log shows a long, multi-stage dropper chain with numerous Sysqem*.exe payloads dropped into the Te mp folder (C:\Users\admin\AppData\Local\Temp) and then further staged executables (e.g., Sysqemrjedk. exe, Sysqemtz. . . etc.).  This pattern is characteristic of Scar's self-propagating worm behavior, where a small dropper copies itself into accessible locations and spawns additional payloads for propagation.
    - Repeated file writes of PE files to the Temp directory during cleanup (FILE_OP_FLAG_WRITTEN_ON_) CLEANUP indicate the drop-and-persist/evade approach typical of Scar-like worms, which survive to later stages by writing new executables to disk.
  - The process chains trace through system processes (smss.exe → winlogon.exe → explorer.exe → multiple Sysqem*.exe processes) and high-integrity, high-network-level activity across the chain.  This aligns with Scar's propagation and persistence through Windows startup/Explorer lineage.
    - Although explicit USB/removable-media activity is not shown, the pervasive local drop-and-run behavior (new Sysqem*.exe created in Temp, then further child processes) matches Scar's described pattern of self-propagation and persistence on connected hosts.
  In short, the observed self-replicating dropper chain in Temp with multi-stage process chains and cleanup-file writes is consistent with the behavioral profile of Worm.Win32.Scar.

---

