# OpenReview forum: "WatchLog: From a Glimpse to Decision—Rapid Event Reasoning in Endpoint Detection and Response Logs with Multimodal LLMs"
_ICLR.cc/2026/Conference — Submitted to ICLR 2026_

### Official Review · Reviewer_HzDW · 2025-10-27

**Soundness:** 3
**Presentation:** 3
**Contribution:** 3
**Rating:** 4
**Confidence:** 3

**Summary:**

The paper introduces WatchLog, a novel framework designed for malicious behavior detection and reasoning in Endpoint Detection and Response (EDR) logs. Traditional EDR systems face challenges in efficiently processing long, complex log sequences and providing interpretable reasoning for security analysts. WatchLog addresses these issues by transforming raw logs into video-structured representations, using a multimodal large language model (LLM) approach. It incorporates a three-stage framework where events are first encoded as images, then temporally aggregated into video-like sequences, and finally fine-tuned to generate both attack family labels and human-interpretable rationales. Experiments on the newly constructed EDR8M-20R dataset show that WatchLog achieves high detection accuracy and recall, while also producing reliable reasoning and significantly improving inference efficiency.

**Strengths:**

1. The approach of transforming raw EDR logs into video-structured representations is highly innovative. By encoding individual log events as images and aggregating them temporally, the model can efficiently process long, high-dimensional sequences and capture both event-level semantics and temporal dynamics.
2. WatchLog outperforms state-of-the-art baselines in both detection accuracy and reasoning quality. It achieves near-perfect binary accuracy (99.8%) and a recall of 100%, while also generating high-quality rationales that are coherent, consistent, and complete.
3. The introduction of a temporal cross-attention mechanism to model long sequences effectively is a significant strength. This mechanism helps preserve the temporal dependencies between log events while reducing the computational cost of processing long logs.

**Weaknesses:**

1. The transformation of logs into video-like structures and the use of large multimodal models introduce significant computational overhead. While the framework reduces redundancy through temporal aggregation, the overall cost could still be high, especially when processing large-scale, real-time logs in a production environment.
2. Although WatchLog demonstrates good performance on the EDR8M-20R dataset, which is large, there is still uncertainty regarding its scalability to even larger datasets or real-time systems where the log sequences are extremely long. The computational bottleneck remains an issue, particularly when handling ultra-long logs typical in enterprise environments.
3. While the paper compares WatchLog with several baselines, it lacks a detailed comparison with other advanced EDR systems. A more thorough analysis against other multimodal or transformer-based models in real-world EDR environments could provide a clearer understanding of the model's strengths and weaknesses.

**Questions:**

Please refer to the weaknesses.

---

> ### Author Response · Authors · 2025-11-20
>
> **Response to Q1**
>
> We appreciate the reviewer’s attention to efficiency, which is indeed a critical factor in large-scale, real-time log analysis. Our framework is specifically designed to maintain high detection performance while reducing computational overhead.
>
> Experimental evidence shows:
>
>    -Performance: As reported in Tables 2–3 (L338–L349, L369–L373), WatchLog consistently outperforms all baseline models across key accuracy metrics.
>
>    -Efficiency: As shown in Table 5 (L409-414), compared with pure LLM-based methods, our approach significantly reduces inference resources and time. For example, relative to the smallest baseline, Qwen2.5-1.5B, processing 1,024K tokens achieves a 50.6% reduction in GPU-MU and a 233.6× speedup in TTFT (from 4.3 min to 1.1 s). These results also indicate that as input size grows, WatchLog achieves increasing relative efficiency, effectively supporting large-scale, real-time log inference.
>
> **Response to Q2**
>
> As mentioned in our response to Question 1, processing ultra-long logs (e.g., 1M-token inputs) is inherently challenging, and existing methods often fail or incur prohibitive costs.
>
> For example, inference with Qwen2.5-1.5B (Table 5, L409-414) consumes up to 98.46 GB of GPU memory and takes nearly 4.3 minutes TTFT. In contrast, WatchLog’s design enables strong detection performance while substantially reducing resource usage: GPU memory drops to ~48.61 GB and TTFT to ~1 s, supporting real-time processing even for ultra-long logs.
>
> For even longer inputs, considering both security event completeness and network transmission latency constraints, 1M tokens represents a practical industrial balance. In engineering implementations, processing can be further accelerated via log segmentation and sliding-window strategies.
>
> **Response to Q3**
>
> We thank the reviewer for the suggestion. Indeed, comparing against a broader set of advanced EDR systems would provide a more comprehensive evaluation. However, industrial-grade EDR systems are typically closed-source or inaccessible, so a full comparison is not currently feasible.
>
> Nonetheless, using open-source academic datasets and methods, our current experiments already demonstrate that WatchLog achieves significant advantages in both detection performance and reasoning efficiency compared with several recent representative baselines, including Transformer-based models (Tables 2, L338–L349).
>
> Furthermore, we commit to publicly releasing our data and code in the future, providing a reproducible benchmark for the academic community and promoting the development of AI methods for EDR log analysis.

---

### Official Review · Reviewer_Nyjr · 2025-10-30

**Soundness:** 2
**Presentation:** 3
**Contribution:** 3
**Rating:** 6
**Confidence:** 3

**Summary:**

```markdown
This paper presents WatchLog, a three-stage multimodal framework that converts structured EDR logs into event-level image-like tensors (via a kvEmbedding), stacks them into a video-like tensor, applies pixelwise temporal cross-attention to compress T events to t condensed frames (t ≪ T), and feeds the resulting compact representations to a multimodal LLM to produce an attack-family label plus a natural-language rationale. The authors introduce the EDR8M-20R dataset and report strong detection and rationale-quality results, claiming large gains over several traditional ML methods and LLM baselines. Training proceeds in three stages: image↔event pretraining, video↔log pretraining, and supervised fine-tuning.
```

**Strengths:**

```markdown
- Novel architectural framing: encoding structured key–value event data into a `c × h × w` tensor and applying spatial+temporal attention is an interesting and original way to reuse video–vision-language machinery for ultra-long log sequences.
- Clear staged training recipe that separates event grounding, temporal aggregation, and supervised reasoning — this is a sensible decomposition.
- Scalability focus: the temporal compression design and empirical profiling (GPU memory/latency) address an important practical challenge for long-context EDR analysis.
- Thorough ablation of some components (temporal compression length, alignment pre-training presence).
```

**Weaknesses:**

```markdown
1. Core conceptual justification: “image/video” framing may be an implementation of token compression rather than a genuine multimodal/visual inductive bias.
   - Concern: Mapping tokens into a ViT hidden-dimension tensor (`c × h × w`) and applying ViT/cross-attention could be equivalent to learned projection + pooling in the NLP domain. Without evidence of visual/structural priors (locality, patch structure) helping, the VLM framing risks being a different parameterization of token compression.
   - Required experiments/analyses:
     - Compare WatchLog to strong non-visual baselines that perform token/event compression with similar compute and parameter budgets:
       - Hierarchical transformer or attention pooling that compresses T events to t vectors, then feeds those to the same LLM.
       - Perceiver/Perceiver IO, Set Transformer, Linformer, BigBird/Longformer sparse-attention adaptations.
     - Ablate spatial layout:
       - Shuffle spatial patch arrangement (randomize patch order) and report performance; if performance is unchanged, spatial layout is not contributing a visual inductive bias.
     - Ablate initialization and pretraining:
       - Try initializing the ViT with visual pretraining (ImageNet) vs random init; if visual pretraining helps, this supports the “visual” claim.
     - `kvEmbedding` ablations:
       - Replace with mean, max, or standard attentive pooling; report impacts.
     - Diagnostics:
       - Visualize attention maps / pixelwise cross-attention across time to show whether spatial channels consistently encode field-level semantics.
     - Comparative scaling curves:
       - Plot performance vs compressed length t for both ViT pipeline and a comparable NLP-compression baseline to demonstrate different scaling behavior.
   - Rationale for asking these: these will clarify if the main contribution is truly exploiting vision-language inductive biases or simply proposing an effective token-compression pipeline implemented with ViT layers. If the latter, authors must reframe claims accordingly.

2. Fairness and interpretability of baseline comparisons — unusually high false alarm rates for traditional baselines on `EDR8M-20R`.
   - Concern: Table 2 shows traditional methods with very high false alarm rates (65–100%). The manuscript does not analyze why these methods degrade so much on `EDR8M-20R`, making it hard to tell if WatchLog’s advantage arises from algorithmic superiority or dataset/experimental artifacts.
   - Required clarifications/experiments:
     - For every baseline, state explicitly whether it was fine-tuned on the same training data or evaluated zero-shot. Provide per-baseline training details (data used, epochs, hyperparameters).
     - Provide ROC and precision–recall curves and AUCs for binary detection baselines, not only single operating points. Describe how decision thresholds were selected (e.g., tuned on validation).
     - Provide per-family confusion matrices for baselines and WatchLog; include representative failure cases (sample logs) where baselines issue false alarms.
     - Sanity checks:
       - Re-run at least one baseline on an established external dataset and show reproduced/expected performance to rule out implementation bugs.
       - If baselines succeed on external benchmarks but fail on `EDR8M-20R`, analyze dataset characteristics (class imbalance, templated artifacts, family similarity to benign events) that cause the drop.
     - Preprocessing parity:
       - Confirm that baselines receive the same input representation (same sampling of events, same field filtering, masking) and any sliding-window segmentation is reported and matched when applicable.

3. Use of GPT-family models for generating training rationales and for automated evaluation introduces circularity and bias.
   - Concern: The authors use an LLM to produce event descriptions, log stories, and SFT rationales (Appendix E) and use GPT-5 as the automated evaluator for rationale quality (Appendix G). This raises two issues:
     - Training and evaluation are potentially correlated: the model may learn to mimic GPT-produced style and therefore score well under a GPT-based evaluator. This is circular and can overestimate genuine interpretability for human analysts.
     - Reliance on GPT for ground-truth rationales needs documentation: were GPT-generated rationales human-verified? If so, by whom and with what inter-annotator agreement?
   - Required clarifications/experiments:
     - Report the fraction of GPT-generated labels/rationales that were human-verified/corrected. Provide details about the human annotator pool (number, qualifications, compensation) and inter-annotator agreement metrics.
     - Add a human expert evaluation (even a small-scale study, e.g., 50–100 samples) of rationale quality, and report agreement with GPT-5 scores.
     - Evaluate rationale quality with at least one automated metric independent of GPT (e.g., ROUGE/L, BERTScore against human-written rationales) and report results.
     - If human verification was not performed, acknowledge limitations and present plans or partial analyses.

4. Dataset release, ethics, and potential dual-use considerations are inadequately detailed.
   - Concern: The paper claims `EDR8M-20R` is publicly released and contains over 8M events and 172 malicious families. Handling malware traces and potentially sensitive logs requires careful redaction, legal review, and a release policy to mitigate misuse.
   - Required clarifications:
     - Provide the exact release plan: what will be shared (raw logs, masked logs, derived features, images), distribution license, access controls (open vs gated), and redaction/sanitization protocols (how PII, hostnames, file paths, sample hashes are sanitized).
     - Provide ethics/institutional approvals (if any) and steps taken to ensure legal compliance.
     - Discuss dual-use risk mitigation (e.g., rate-limited access, licensing with restrictions for offensive use, vetting requester institutions).
     - Clarify whether the “expert-verified” labels are produced entirely from virtualized attack scripts or include any captures from real enterprise environments.

5. Reproducibility and statistical reporting
   - Concern: Several headline numbers (e.g., 100% recall, 99.8% binary accuracy) are very high and may indicate leakage or evaluation artifacts.
   - Required clarifications:
     - Report standard deviations across multiple runs (3–5 seeds) for key metrics. Provide confidence intervals or p-values for comparisons to baselines.
     - Release a simple reproducibility checklist and a single table listing all hyperparameters, pretraining steps, number of examples used per step, and total compute used for each stage.

6. Robustness and threat model
   - Concern: The authors do not test how robust their image/video encoding is to simple manipulations that an adversary could produce (e.g., adding benign noise events, truncating or reordering events, obfuscating key–value fields).
   - Required experiments:
     - Simple evasion tests: add benign noise events, random field swaps, or reorder events while preserving counts; report detection and rationale quality changes.
     - Permutation test: permute temporal ordering to evaluate whether temporal cross-attention is leveraging order-sensitive signals.
     - Adversarial perturbation: assess whether small edits to critical fields drastically change outcomes.

Minor / presentation suggestions
- Add per-family precision/recall and a confusion matrix for the main test set in the main paper (not only in appendix).
- Combine dataset and preprocessing details into a single, clear subsection (number of hosts, number of runs, how train/test splits were constructed to avoid leakage).
- Provide several negative examples: cases where WatchLog mislabels or provides incorrect/misleading rationales, with analysis of why.
- Provide the exact prompts used for GPT tasks in an appendix (some prompts are present, but ensure full prompts and examples are included).
```

**Questions:**

```markdown
1. Dataset and release:
   - Will `EDR8M-20R` be publicly released as raw logs, processed images, or only labels and summary statistics? If not fully public, can a vetted subset be released for reproducibility?
2. GPT-produced rationales:
   - Were GPT-generated event descriptions/log stories/rationales human-verified? If yes, what fraction and what was the inter-annotator agreement?
3. Baseline protocol:
   - For each baseline in Table 2, list whether it was fine-tuned on your training data or evaluated zero-shot. Provide hyperparameters/epochs used.
4. Visual priors:
   - Did you try visual pretraining for the ViT backbone (e.g., ImageNet) and compare to random init? Does visual pretraining meaningfully change results?
5. Spatial arrangement sensitivity:
   - What happens if the patch layout (`M = h×w`) is permuted across training and testing? Does performance drop?
6. Robustness:
   - Have you measured how small, structured perturbations (noise events, reordering, masking key fields) impact detection and rationale quality?
```

**Details Of Ethics Concerns:**

```markdown
- Dual-use: the dataset plus explicit prompt recipes and model code could be misused by attackers to learn which behavioral signals are used for detection. The authors should adopt gated release, redaction, or responsible-disclosure policies.
- Dataset privacy: need explicit confirmation that no PII survived redaction (usernames, real hostnames, directories, other identifiers) and that legal approvals were obtained.
- Label provenance: the heavy use of GPT for generating rationales and event descriptions must be documented; reliance on a single proprietary LLM for both training data generation and evaluation introduces systemic bias that should be acknowledged and mitigated with human verification.
```

---

> ### Author Response · Authors · 2025-11-20
>
> **Response to Q1**
>
> We commit to publicly releasing the EDR8M-20R dataset upon paper acceptance, including: anonymized raw logs, processed visual representations, dataset statistics, and all relevant code. This release will enable the community to fully reproduce and validate our results.
>
> **Response to Q2**
>
> We sincerely appreciate the reviewer’s insightful comments.
>
> All GPT-produced event descriptions, log stories, and rationales in both pre-training and SFT phases were subjected to strict human expert auditing. We removed any samples with missing key information, factual errors, or invalid reasoning chains. Consequently, the final pre-training and SFT datasets consist exclusively of human-verified, high-quality annotations.
>
> **Response to Q3**
>
> Appendix F (L756–802) details the experimental protocols for all baselines. Specifically:
>
>    -Fine-tuned on our training data: LGTF, ADE, DrSec, Qwen2.5-1.5B, Qwen2.5-7B, LLaMA3.1-8B
>
>    -Zero-shot evaluation: DeepSeek-R1-671B, Qwen3-235B-A22B
>
> Complete hyperparameters, training epochs, and configuration details are provided in Appendix F (L756–802). We commit to publishing all code and configurations used to reproduce baseline results alongside our datasets and framework code.
>
> **Response to Q4**
>
> The visual backbone we used was pretrained. To assess the impact of random initialization, Table 6 (L426-431) reports a similar ablation study comparing models with and without aligned visual pretraining. The results show that aligned pretraining substantially improves performance, highlighting its importance for effective multimodal reasoning.
>
> **Response to Q5**
>
> Our method does not assume a fixed patch layout. To directly address the reviewer’s concern, we will conduct experiments with randomly permuted patch layouts during training and testing. We will report results in the revised manuscript before November 30.
>
> **Response to Q6**
>
> Robustness to noise and event reordering is already considered in our design. Figure 3 (b) (L459-470) evaluates performance when the number of events is increased randomly at test time, simulating both noisy events and reordered sequences.
>
> Additionally, in EDR analysis scenarios, some malicious behaviors rely entirely on specific fields or events (e.g., detecting a USB-borne malware infection becomes impossible if the event_type field is masked); therefore, experiments that mask key fields generally provides little practical insight, as it disrupts the core semantics essential for forensic reasoning.
>
> **Response to Q7**
>
> We sincerely appreciate the reviewer’s thorough considerations and concerns.
>
> Regarding Dual-use and Dataset Privacy: these issues were anticipated in our study. The dataset release will undergo review by our institutional ethics committee and will follow a gated access policy, limited to research purposes and adhering to responsible disclosure guidelines. At the same time, to preserve log semantics and analytical value, identifiable information (e.g., paths, usernames, hostnames, directories) is anonymized using encryption techniques. This process replaces only the sensitive identifiers while preserving the semantic patterns and statistical distribution of the logs, preventing attackers from bypassing detection and ensuring both privacy and reproducibility of all analyses.
>
> Regarding Label Provenance: we conducted a cross-model consistency analysis using two additional strong evaluators: Gemini-2.5 Pro (closed-source) and Qwen3-235B-A22B (open-source). The results show highly consistent scoring trends across all evaluators, confirming that the Reasoning Quality assessment is robust and not tied to a single model family.
>
> | | Coh.↑| Cons.↑| Comp.↑|
> |-|:-:|-:|:-:|
> | GPT-5| 91.1| 99.0| 71.7|
> | Gemini-2.5 Pro| 90.9| 87.0| 73.3|
> | Qwen3-235B-A22B| 99.0| 99.1| 60.5|
>
> To strengthen transparency and rigor, all reasoning-related metrics in the revised manuscript are now reported as mean values. (Table 2, L346–L349; Table 3, L369–L373; Table 6, L429–L431; Figure 3, L450-470).

---

> ### Comment · Reviewer_Nyjr · 2025-11-25
>
> I have read the responses and other reviewers' comments. Some of my concerns are addressed. I tend to keep my score.

---

> > ### Author Response · Authors · 2025-11-26
> >
> > We sincerely appreciate the reviewer’s positive evaluation of our work. As promised in our response to Q5, we have conducted additional experiments with randomly permuted patch layouts during training and testing. Specifically, we carried out two settings: (1) training with randomly permuted patches while testing with the default layout (alphabetically by key), and (2) training and testing both with randomly permuted patches, where we performed five test runs and reported the average results.
> >
> > The results indicate that random permutation substantially reduces the false alarm rate, albeit with a slight decrease in overall accuracy. Regarding the reasoning-quality metrics, applying random permutation only during training leads to improvements in Coh. and Cons., with a modest drop in Comp.; when random permutation is applied to both training and testing, the decrease in reasoning quality remains within acceptable bounds. These findings further demonstrate the robustness of our method to variations in spatial layout. All supplementary experiments and the corresponding discussions have been incorporated into the revised manuscript (L503-525).
> >
> > | |FA↓|OA↑| Coh.↑| Cons.↑| Comp.↑|
> > |------|:----:|-----:|:----:|:----:|:----:|
> > | WatchLog-2B (default)| 5.0| 90.3| 93.6| 95.0| 68.5|
> > | WatchLog-2B (perm. train only)| 0.0| 88.5| 94.2| 96.2| 67.1|
> > | WatchLog-2B (perm. train \& test)| 1.0| 88.0| 92.9| 94.6| 64.8|
> >
> > We sincerely appreciate the reviewer’s follow-up assessment and are grateful that several of your concerns have been addressed. We are also encouraged by your recognition of the strengths of our work — including the novel architectural framing for structured event data, the clear staged training pipeline, the scalability-oriented temporal compression design, and the thorough ablations. Your acknowledgment of these aspects is highly motivating for us.
> >
> > If there remain any unresolved points or aspects that could benefit from further clarification, we would be more than happy to provide additional analyses or explanations. This work is of great importance to us, and your recognition would be extremely valuable. We genuinely hope that the clarifications and new experiments presented in the revised version may encourage you to reconsider your evaluation in a more positive light.

---

### Official Review · Reviewer_cDqe · 2025-10-31

**Soundness:** 4
**Presentation:** 3
**Contribution:** 3
**Rating:** 6
**Confidence:** 3

**Summary:**

The paper transfer the tradition EDR problem to a video understanding task, with the help of MLLMs, they achieve more accurate and efficent detection.

**Strengths:**

1. It's the first work that converts raw logs from the EDR system into video-structured inputs and utilize MLLM for end2end reasoning.
2. To solve the complex temporal relationship in log data, they add a temporal cross-attention block into Qwen2VL after the replaced kvEmbedding layer, further squeeze the tmporal information than the original Qwen model.
3. A solid Three-Stage Progressive Training, ensure the image-event alignment,  cross-Event (temporal) dependencies capturing and standard reasoing learning (by SFT).
4. good performance on their own test set and the Carbon Black Cloud subset of ATLASv2.

**Weaknesses:**

1. Data is derived from virtualized enterprise Windows environments, predefined attack scripts, and Windows kernel callback collection. Performance on other platforms and in different scenarios requires further validation.
2. GPT is used throughout the process of generating training data, alignment, and evaluation, raising questions about objectivity and reproducibility.
3. If WatchLog's explicit temporal compression proves advantageous, comparative experiments should be conducted on standard video models to validate its superiority over conventional video backbones.
4. Only test on their own dataset and Carbon Black Cloud subset of ATLASv2. Although it performs well on its test dataset, its generalizability remains questionable.

Also please check questions for detailed weakness explaination.

**Questions:**

1. Pre-training first uses GPT to generate event descriptions for alignment, which injects synthetic text styles and biases into the training distribution. The rationales during the SFT phase are also generated by GPT-5, and the final Reasoning Quality assessment is also scored by GPT-5. **Does this affect objectivity and evaluation accuracy?** Could you report the Reasoning Quality score using another strong model like Gemini-2.5 pro.
2. You can already convert log data into visual data (images and videos). **Why not try smaller non-LLM models (e.g. Visual Classifier: image-ViT, video - VideoMAE/I3D) for direct detection?** This is now changed to a classification task, you can just train a classification head to test, even considering the parameter scale advantage of LLMs, there are now large encoders like InternVL2-ViT-26B available.
3. If your goal contains leveraging the reasoning capabilities of LLM, I believe **it is essential to present the reasoning performance results of large-scale VLMs**. Specifically, in Table 2, you should include the results for Deepseek 671B and Qwen3 235B, even if the zero-shot model's reasoning quality is not good enough, their results may also surpass the 7/8B SFT model.
4. Why did you evaluate only on the Carbon Black Cloud subset of ATLASv2 for external validation? Do you have results on other ATLASv2 subsets or on the full dataset, and **could you evaluate on more public benchmarks?** As it stands, the 66.7% binary accuracy is relatively weak as evidence.
Your internal data contain LLM-generated textual supervision; on external public datasets that lack such text, does the model’s performance degrade?

---

> ### Author Response · Authors · 2025-11-20
>
> **Response to Q1**
>
> We thank the reviewer for raising this important concern.
>
> On GPT-generated annotations: All GPT-generated event descriptions (pre-training) and rationales (SFT) underwent strict human expert auditing. We systematically filtered out samples containing missing information, factual inconsistencies, or invalid reasoning chains. As a result, the final training corpus consists only of human-validated, high-fidelity annotations, ensuring that the retained synthetic data is reliable and semantically accurate.
>
> On objectivity of Reasoning Quality evaluation: To directly address the concern regarding single-model evaluation, we performed cross-model consistency analysis using two additional strong evaluators: Gemini2.5 Pro (closed-source) and Qwen3-235B-A22B (open-source). All evaluators produced highly consistent trends across Coherence, Consistency, and Completeness, suggesting that the Reasoning Quality scores are not tied to a specific model family.
>
> | | Coh.↑| Cons.↑| Comp.↑|
> |-|:-:|-:|:-:|
> | GPT-5| 91.1| 99.0| 71.7|
> | Gemini-2.5 Pro| 90.9| 87.0| 73.3|
> | Qwen3-235B-A22B| 99.0| 99.1| 60.5|
>
> To further improve transparency, the revised manuscript now reports each reasoning metric as the mean score across the three evaluators (Table 2, L346–L349; Table 3, L369–L373; Table 6, L429–L431; Figure 3, L450-470).
>
> **Response to Q2**
>
> We appreciate the reviewer’s suggestions.
>
> On why we do not directly use pure vision models: a) A core motivation of our work is meeting real-world security needs for explainability. Beyond accuracy, security analysts need interpretable evidence—reasoning traces, anomaly indicators, and behavior-level explanations—for forensic decision-making. b) Pure vision models cast the task as classification, losing the semantic reasoning advantages of LLMs, resulting in limited interpretability and weaker generalization to unseen attacks. c) By first converting logs into visual representations and then performing multimodal reasoning, our model emulates the human process of visually examining information and then interpreting its semantics before reasoning over it. Similar multimodal strategies for ultra-long text have also appeared in recent works such as DeepSeek-OCR[1] and Glyph[2] (both published after ours), supporting the soundness and practicality of our design.
>
> On visual classification baselines: Nevertheless, We also implemented a VideoMAE classifier on our log-to-visual representation and extensively tuned hyperparameters, but it failed to converge (the implementation will be released with the code). This is likely due to pure vision models like VideoMAE being inherently misaligned with our “visual data”and struggling to learn meaningful representations, similar to how LSTMs and CNNs are suited to different data structures and learning objectives. In contrast, multimodal VLMs are natively equipped with jointly trained visual-language spaces, making them naturally better suited to this task. Notably, our current event-converted architecture proved the most effective among the design variants we tested.
>
> [1]Wei, Haoran, Yaofeng Sun, and Yukun Li. "DeepSeek-OCR: Contexts Optical Compression." (2025).
>
> [2]Cheng, Jiale, et al. "Glyph: Scaling Context Windows via Visual-Text Compression." (2025).
>
> **Response to Q3**
>
> Thank you for the constructive suggestions.
>
> On large LLM reasoning performance: Table 2 already reports zero-shot results for DeepSeek-R1-671B and Qwen3-235B-A22B. Both underperform compared to the 7B/8B SFT models, likely due to the extreme scarcity of industrial EDR logs in general pre-training corpora, which limits zero-shot alignment to this highly specialized domain.
>
> On larger VLM evaluations: We will conduct additional experiments using Qwen2.5-VL-3B-Instruct and Qwen2.5-VL-7B-Instruct to systematically assess reasoning performance across model scales. The results and discussion will be provided and included in the revised manuscript before November 30.
>
> **Response to Q4**
>
> We appreciate the reviewer’s attention to external validation.
>
> On use of Carbon Black Cloud subset: The Carbon Black Cloud subset is the only ATLASv2 component that reflects realistic EDR telemetry; other subsets (e.g., Firefox browsing logs, Wireshark DNS traces) diverge substantially from EDR contexts.
> On more public benchmarks: After reviewing existing resources and consulting with security experts, we found no other publicly accessible datasets that contain EDR-style logs suitable for meaningful evaluation. Nevertheless, we will continue monitoring the field and plan to expand evaluations in future work.
>
> On LLM-generated textual supervision: Textual supervision is used only during training. At inference, the model operates solely on raw log data. Table 4 (L390–396) shows that our model remains strong performance on external EDR data, confirming robust generalization.
>
> Finally, we commit to releasing all datasets and code to ensure full transparency and reproducibility.

---

> > ### Comment · Reviewer_cDqe · 2025-11-20
> > **Response to authors' rebuttal.**
> >
> > You have mostly addressed my concerns, and I will maintain my positive score for acceptance.

---

> > > ### Author Response · Authors · 2025-11-26
> > >
> > > We sincerely thank the reviewer for the positive assessment of our work. As committed in Response to Q3, we have completed additional evaluations using Qwen2.5-VL-3B-Instruct and Qwen2.5-VL-7B-Instruct as the backbones of our proposed WatchLog framework. The results show that our method consistently improves reasoning performance across model scales, confirming that WatchLog generalizes well to stronger VLM backbones and can effectively leverage their enhanced capacity. Detailed results and analysis have been incorporated into the revised manuscript (L481-502)
> > >
> > > | | FA↓|OA↑|Coh.↑| Cons.↑| Comp.↑|
> > > |------|:----:|-----:|:----:|:----:|:----:|
> > > |WatchLog-2B| 5.0| 90.3| 93.6| 95.0| 68.5|
> > > |WatchLog-3B|0.0| 91.3| 95.8| 96.6| 74.4|
> > > |WatchLog-7B|0.0| 92.5| 96.3| 97.3| 78.2|
> > >
> > > We sincerely appreciate the reviewer’s recognition of the key contributions of our work — including the conversion of raw EDR logs into video-structured inputs for end-to-end MLLM reasoning, the introduction of temporal cross-attention to model complex event dependencies, the three-stage progressive training pipeline, and the strong empirical results on both our benchmark and the ATLASv2 subset.
> > >
> > > Your acknowledgment of these strengths is highly encouraging to us. Should there be any remaining aspects that could benefit from further clarification or additional analysis, we would be glad to elaborate. This work is of great importance to us, and your recognition would be extremely valuable. We genuinely hope that the new experiments and revisions may encourage you to reconsider your evaluation in a more positive light.

---

### Meta-Review · Area_Chair_DsWF · 2026-01-09

**Summary:**

The paper proposes WatchLog, an innovative method that transforms EDR logs into video-like representations for processing by large multimodal models, achieving high detection accuracy and reasoning quality. However, reviewers raised critical concerns about the core conceptual justification (whether the video framing offers genuine visual priors or is merely token compression), potential circularity and bias from extensive GPT use in training and evaluation, insufficient validation of generalizability and baselines, unclear dataset release and ethics, and inadequate robustness and reproducibility reporting.

**Reviewer Concerns:**

The rebuttal may partially address concerns about baseline implementation details and dataset release plans by providing additional clarifications and promised future work. However, some significant conceptual concerns, e.g., the inherent circularity and potential bias from using GPT for both training data generation and final evaluation, remain fundamentally unresolved

**Reviewer Scores:**

After the rebuttal, the scores remained hovering around the average, accompanied by a reject opinion. Given the unresolved and inadequately explained issues, e.g., potential bias from using GPT for both training data generation and final evaluation, the reviewers generally are unlikely to strongly support acceptance of this paper. Based on these discussions and given the current high volume of submissions, the AC (Area Chair) leans towards rejection.

---

### Decision · Program_Chairs · 2026-01-26

Reject